# FLOW MAP LEARNING VIA NONGRADIENT VECTOR FLOW

**Mark Goldstein**[1]***, Anshuk Uppal**[2]**, Raghav Singhal**[3]**, Aahlad Puli**[3]**, & Rajesh Ranganath**[3]
[1]Center for Computational Mathematics, Flatiron Institute.
[2]Department of Applied Mathematics and Computer Science, Technical University of Denmark.
[3]Courant Institute, New York University.

## ABSTRACT

Diffusion and flow-based models benefit from simple regression losses, but inference (i.e, producing samples) incurs significant computational overhead because it requires integration. Consistency models address this overhead by directly learning the flow maps along the ODE trajectory, revealing a design space *for the learning problem* between one-step and many-step approaches. However, existing consistency training methods feature computational challenges such as requiring model inverses or backpropagation through iterated model calls, and do not always prove that the desired ODE flow map is a solution to the loss. We introduce SGFlow, an approach for learning flow maps that bypasses explicit invertibility constraints and expensive differentiation through model iteration. SGFlow trains a model to compute both the ODE solutions and the implied velocity from scratch by following non-conservative dynamics with a stationary point at the desired flow map. On the CIFAR image benchmark, SGFlow attains a favorable relationship of FID to step count, relative to flow matching, MeanFlow, and several other flow map learning methods.

## 1 INTRODUCTION

Diffusion and flow models (Sohl-Dickstein et al., 2015; Ho et al., 2020; Song et al., 2020; Kingma et al., 2021; Albergo and Vanden-Eijnden, 2022; Singhal et al., 2023; Pandey and Mandt, 2023; Bartosh et al., 2024; Singhal et al., 2024; Albergo et al., 2023; Lipman et al., 2022; Liu et al., 2022) have improved generation in domains such as proteins (Abramson et al., 2024) and images (Peebles and Xie, 2023; Esser et al., 2024). Sampling from these models typically requires numerically integrating an ordinary or stochastic differential equation. Numerical integration requires multiple forward passes of a neural network, leading to increased sampling latency and cost.

To ameliorate this generation cost by changing the training, recent approaches for consistency modeling and map matching (Song et al., 2023; Song and Dhariwal, 2023; Kim et al., 2023; Lu and Song, 2024; Boffi et al., 2024; 2025) aim to learn direct mappings from noise to intermediate or final data points along trajectories defined by probability flow ODEs, thereby avoiding costly integration. However, the methods have their respective complexities. For example, flow map matching requires model invertibility, while consistency models need either to map in one step or introduce extra steps that leave the target ODE trajectory.

We introduce SGFlow (for StopGrad Flow), an approach that builds on flows and map matching methods and

- Has true flow map as a unique stationary point

- Does not restrict the class of neural networks used (e.g., to invertible functions)

- Does not require auxiliary losses involving invertibility or adversarial optimization

- Does not require optimizing through nested calls to the model

- Allows for generation along the ODE trajectory with any number of steps

---

*mgoldstein@flatironinstitute.org

| Methods | Multistep | Follows ODE | Sim. Free | Regression Loss | Inverse Required | Prove Optimum | Model Nesting |
|---|---|---|---|---|---|---|---|
| Consistency Distillation (Song et al., 2023) | ✗ | ✓ | ✗ | ✓ | ✗ | ✓ | ✗ |
| Consistency Training (Song et al., 2023) | ✗ | ✓ | ✓ | ✓ | ✗ | ✓ | ✗ |
| Consistency Trajectory Models (Kim et al., 2023) | ✓ | ✓ | ✗ | ✗ | ✗ | ✓ | ✗ |
| L-FMM (Boffi et al., 2024) | ✓ | ✓ | ✓ | ✓ | ✓ | ✓ | ✗ |
| LSD (Boffi et al., 2025) | ✓ | ✓ | ✓ | ✓ | ✗ | ✓ | ✓ |
| ESD (Boffi et al., 2025) | ✓ | ✓ | ✓ | ✓ | ✗ | ✓ | ✗ |
| PSD (Boffi et al., 2025) | ✓ | ✓ | ✓ | ✓ | ✗ | ✓ | ✓ |
| MeanFlow (Geng et al., 2025) | ✓ | ✓ | ✓ | ✓ | ✗ | ✗ | ✗ |
| **SGFlow (this work)** | ✓ | ✓ | ✓ | ✓ | ✗ | ✓ | ✗ |

Table 1: **Comparison to prior works**. We categorize consistency modeling (flow map learning) techniques and our proposed SGFlow method according to: (1) ability to adjust sampling steps post-training, (2) whether they follow the PF-ODE (Song et al., 2020), (3) whether they allow simulation-free training, (4) whether their objectives use regression, (5) whether model inversion is required during training, (6) whether the true flow map is proven to be optimal or stationary, and (7) whether differentiation passes through nested model calls See Section 5 for details.

Existing methods for learning flow maps fall into a few categories in terms of their challenges; all challenges relate to the idea that a flow map is characterized by certain derivative properties and that losses minimize squared error to make these properties hold. Flow map matching and related methods rely on a fundamental relationship between invertible mappings and ordinary differential equations (ODEs). This relationship typically requires explicitly computing both the forward map (the model being trained) and its inverse during training, complicating training, and requires explicitly materializing a large derivative matrix in the forward pass. Other methods, such as MeanFlow (Geng et al., 2025), do not explicitly enforce the model inverse identities, but do not prove that their loss is minimized at the true flow map or that the optimization is stationary at that solution.

SGFlow avoids the complexity of tracking a model and its inverse, as well as materializing derivatives, by exploiting an alternate identity involving only Jacobian-vector products (JVPs) without inverse functions. This identity allows us to formulate the objective purely in terms of the forward map, without needing explicit access to its inverse. Since solutions to ODEs naturally produce invertible mappings, the SGFlow objective implicitly encourages invertibility without explicitly enforcing it. Thus, at optimality, SGFlow yields a continuously differentiable function that precisely integrates the velocity field, directly generating the desired data distribution. We summarize the trade-offs among recent methods in Table 1 and in Section 5.

Experimentally, for a basic training setup using the same common architecture, we ask how flow matching, MeanFlow, SGFlow, Lagrangian Map Matching, Eulerian Map Matching, and Progressive Map Matching compare in moderate dimensions (CIFAR-10) on unconditional metrics (FID) when decreasing the number of sampling steps. We additionally compare GPU memory usage for these methods.

## 2 BACKGROUND

Stochastic interpolants (Lipman et al., 2022; Albergo et al., 2023), and more generally most diffusion and flow methods, hereafter just *flows* pose generative modeling as transport of a simple base density to a target density. Interpolants tackle the problem as follows. For $t \in [0, 1]$:

1. Choose $(\alpha_t, \sigma_t)$ where $\alpha_0 = \sigma_1 = 1$ and $\alpha_1 = \sigma_0 = 0$. Commonly, $\alpha_t = 1 - t$ and $\sigma_t = t$.
2. Define $X_t = \alpha_t X_0 + \sigma_t X_1$ for base density $X_0 \sim q_0$ and data $X_1 \sim q_1$ (or vice versa).
3. Learn to produce new samples along the trajectory of densities.

For a function $f$, let $\dot{f}_t := \frac{d}{dt} f_t$. Thus $\dot{X}_t := \dot{\alpha}_t X_0 + \dot{\sigma}_t X_1$. It follows that $X_t$ has density $q_t$ satisfying:

$$\partial_t q_t(x) = -\nabla_x \cdot (q_t(x) v(t, x)), \qquad v(t, x) := \mathbb{E}[\dot{X}_t \mid X_t = x], \qquad (1)$$

where $v$ is called the velocity. The PDE in Equation (1) is derived in the above works. To accomplish step three, one starts by making the observation that a density satisfies Equation (1) if and only if it is the density of the solution to the *probability flow ODE* $dx = v dt$ integrated forward from

$X_0 \sim q_0$ or in reverse from $X_1 \sim q_1$ (Albergo and Vanden-Eijnden, 2024). One then proceeds by first approximating $v$ using the following (simulation-free) loss:

$$\mathcal{L}_v(v_\theta) = \mathbb{E}\Big[\|v_\theta(t, X_t) - (\dot{\alpha}_t X_0 + \dot{\sigma}_t X_1)\|^2\Big]_{X_t = \alpha_t X_0 + \sigma_t X_1}, \tag{2}$$

which has minimizer $v_\theta = v$ and then solving $dx = v_\theta dt$.

**Background on Consistency Methods.** Sampling from flows requires integration where each integration step evaluates a neural network $v_\theta$ modeling a score, velocity, or similar. Therefore, knowing integrals of $v$ or similar quantities directly could, in principle, speed up sampling. The goal of consistency and map matching methods is to learn to map along the trajectory implied by the optimal $v$. We review an example here, with others described in Section 5. Song et al. (2023); Song and Dhariwal (2023) seek to learn a mapping $\hat{g}$ that maps interpolant samples $X_t \sim q_t$ to $\widehat{X}_0$, the $t = 0$ solution to $dx = vdt$ when starting at $X_t$ (note that $\widehat{X}_0$ usually differs from the independent endpoint sample $X_0$ used to draw $X_t$ under the interpolant). The loss measures the distance between modeled outputs when evaluated at two different nearby points. Let $\text{sg}[\hat{g}]$ indicate stopgrad (i.e., bookkeeping a term as a constant when computing loss gradients). Then the loss is:

$$\text{Consistency}(\hat{g}) := \mathbb{E}_{q(X_t)}[\text{dist}(\hat{g}(t, X_t), \text{sg}[\hat{g}](t - \Delta t, \widehat{X}_{t-\Delta t}))]. \tag{3}$$

The sample $\widehat{X}_{t-\Delta t}$ used in the target should optimally come from integrating the true velocity or an approximation $v_\theta$ a small step $\Delta t$ from $X_t$, where $v_\theta$ either comes from a pretrained diffusion model, or $(v_\theta, \hat{g})$ are derived from one another. In practice, further approximations are used to compute $x_{t-\Delta t}$. Approximations are introduced because $\widehat{X}_{t-\Delta t}$ is not simply defined by drawing a second interpolant sample at a smaller noise level, but instead corresponds to integrating the velocity one step from $X_t$; the velocity is unknown and thus may come from a pre-trained model, which increases training cost and may not be a good approximation in the first place.

It is challenging to directly learn solutions in just one step. While this allows multistep sampling, the re-noising step necessarily takes the trajectory off the probability-flow ODE and the resulting updates no longer correspond to integrating the PF-ODE solution. Highlighting the issue with methods featuring 1 time argument, Kim et al. (2023) note that this CM multistep sampling approach "exhibits degrading sample quality with increasing NFE, lacking a clear trade-off between computational budget (NFE) and sample fidelity". In practice, a number of training-time or inference-time changes are made to this setup to try to break apart the problem into somewhere between 1 step and the hundreds of steps used by diffusions (Song et al., 2023; Lu and Song, 2024; Kim et al., 2023; Boffi et al., 2024; Sabour et al., 2025; Geng et al., 2025; Zhou et al., 2025). We discuss the various solutions and their trade-offs in Section 5.

## 3 METHOD

We present SGFlow, a method for learning to solve the probability flow ODE without adversarial training, without model inverse during training, without representing explicit derivative matrices, and without costly simulations from pretrained models. SGFlow trains a model to compute both the ODE solutions and the implied velocity from scratch by following non-conservative dynamics.

Consider a two-time map $f$ that for $t \leq u$ brings $X_t$ up to $x_u$ by solving the probability flow ODE $dx = vdt$. Such an $f$ that integrates $v$ can be defined as follows:

$$f(t, u, x) = x + \int_t^u v(s, X_s)ds = x + \int_t^u v(s, f(t, s, x))ds \tag{4}$$

Differentiating the recursive form on the RHS w.r.t. $t$ using the total (material) derivative yields:

$$\partial_t f + (\partial_x f)v(t, x) = 0, \quad f(u, u, x) = x \tag{5}$$

This is uniquely solved at the true flow map $f$. We can square the left-hand side for a parameterized $f_\theta$ and take an expectation over $X_t$

$$L = \mathbb{E}_{X_t}[\|\partial_t f_\theta + (\partial_x f_\theta)\mathbb{E}[\dot{X}_t|X_t]\|^2] \tag{6}$$

The true map $f$ is the unique minimizer of this loss. Using $v(t, x) = \mathbb{E}[\dot{X}_t|X_t]$, we can expand,

$$L = \mathbb{E}_{X_t}\Big[\|\partial_t f_\theta + (\partial_x f_\theta)\dot{X}_t\|^2 - \|(\partial_x f_\theta)(\dot{X}_t - \mathbb{E}[\dot{X}_t|X_t])\|^2\Big] \tag{7}$$

We can then use the parameterization $f_\theta = x + (u - t)\tilde{f}_\theta(t, u, x)$ for an underlying model $\tilde{f}_\theta$. The parameterization yields two useful properties:

- time derivative: $\partial_t f_\theta(t, t, x) = -\tilde{f}_\theta(t, t, x)$
- Jacobian: $\partial_x f_\theta(t, t, x) = I$

Using these properties and evaluating at $t = u$, we see that the minimization of eq. (7) reduces to flow matching where $\tilde{f}_\theta(t, t, x)$ is trained to match the velocity:

$$L\Big|_{t=u} = \mathbb{E}_{X_t}[\|\tilde{f}_\theta(t, t, x) - \dot{X}_t\|^2], \tag{8}$$

which reveals that for the true $f$, we have that

$$-\partial_t f(t, t, \cdot) = \tilde{f}(t, t, \cdot) = v(t, x) = \mathbb{E}[\dot{X}_t | X_t = x] \tag{9}$$

This motivates replacing the unknown $v$ in eq. (7) with stopgrad$[\tilde{f}_\theta(t, t, \cdot)]$. The stopgrad is used under the principle that since the original $v$ did not provide gradient updates for $f$, neither should a term that approximates it. The **SGFlow** method follows the negative gradient with respect to $\theta$ of:

$$L_{\text{sg}} := \mathbb{E}\Big[\|(\partial_t f_\theta)_{(t,u,X_t)} + (\partial_x f_\theta)_{(t,u,X_t)}\dot{X}_t\|^2 - \|(\partial_x f_\theta)_{(t,u,X_t)}(\dot{X}_t - \text{sg}[\tilde{f}_\theta]_{(t,t,X_t)})\|^2\Big], \tag{10}$$

where sg() means stopgrad() and $f_\theta(t, u, x) := x + (u - t)\tilde{f}_\theta(t, u, x)$. The expectation is taken over $X_t$ sampled by drawing data $X_1$, noise $X_0$, and computing $X_t = \alpha_t X_0 + \sigma_t X_1$ and $\dot{X}_t = \dot{\alpha}_t X_0 + \dot{\sigma}_t X_1$.

In practice, the PDE must hold for all pairs $t \leq u$. Let $q(t, u)$ be a joint distribution with support over $t \leq u$ and with positive probability on $t = u$. Take expectations over time and define $\mathcal{L} = \mathbb{E}_{q(t,u)}[L]$ and $\mathcal{L}_{\text{sg}} = \mathbb{E}_{q(t,u)}[L_{\text{sg}}]$. We now connect $\mathcal{L}$ and $\mathcal{L}_{\text{sg}}$ formally. Theorem 1 shows that optimization of $\mathcal{L}$ and $\mathcal{L}_{\text{sg}}$ stop at the same solutions.

**Theorem 1.** *Let $q(t, u)$ be a joint distribution with support over $t \leq u$ and with positive probability on $t = u$. Let the family $\tilde{\mathcal{F}}$ include functions $\tilde{f}$ that are continuously differentiable in all arguments. Let $X_t = \alpha_t X_0 + \sigma_t X_1$ and $\dot{X}_t = \dot{\alpha}_t X_0 + \dot{\sigma}_t X_1$. Define $f(t, u, x) := x + (u - t)\tilde{f}(t, u, x)$. Take expectations over $q(X_0)q(X_1)$. Let sg stand for stop-gradient. Define $\mathcal{L} = \mathbb{E}_{q(t,u)}[L]$ and $\mathcal{L}_{\text{sg}} = \mathbb{E}_{q(t,u)}[L_{\text{sg}}]$. Then $\tilde{f}^*$ is a stationary point of $\mathcal{L}_{\text{sg}}$ w.r.t. $\tilde{\mathcal{F}}$ if and only if $\tilde{f}^*$ is a stationary point of $\mathcal{L}$ w.r.t. $\tilde{\mathcal{F}}$.*

This is shown in Section A.5.

**Intuition.** The main point of the theorem is to establish that $\mathcal{L}_{\text{sg}}$ has the same set of solutions as $L$ despite not having access to $v$. The intuition is that, despite the stopgrad, when $t = u$, $\mathcal{L}_{\text{sg}}$ tries to match the velocity. We show that $\mathcal{L}_{\text{sg}}$ is not at a stationary point when this velocity estimate is inaccurate, so the optimization continues moving and does not become stuck at functions that distill an incorrect velocity. As this match improves so does the match between the parameter updates from $\mathcal{L}_{\text{sg}}$ and $\mathcal{L}$ at $t \neq u$. The main reason this works is that $\tilde{f}(t, t, \cdot)$ appears in other terms outside of the stopgrad, and those terms tell it where to go. This is crucial and not all stopgrad optimizations benefit from this property.

**Computation.** Both terms in $\mathcal{L}_{\text{sg}}$ can be computed as expected squared norms of Jacobian-vector products (JVPs), which use forward-mode autodifferentiation to avoid explicitly materializing Jacobians, saving memory. Using PyTorch notation,

$$\text{JVP}[f, (t, u, x), (a, b, c)] := (\partial_t f) \cdot a + (\partial_u f) \cdot b + (\partial_x f) \cdot c$$

for $(\partial_t f, \partial_u f, \partial_x f)$ evaluated at $(t, u, x)$. For the first loss term, $a = 1$, $b = 0$ and $c = \dot{X}_t$ and for the second loss term, $a = 0$, $b = 0$, and $c = \dot{X}_t + \text{sg}[\partial_t f(t, t, X_t)] = \dot{X}_t - \text{sg}[\tilde{f}(t, t, X_t)]$. Though we have two distinct JVPs, we can split the batch and randomly assign either pair of $(a, c)$ values to each batch element.

**Nongradient Flow** Following the update rules of $\mathcal{L}_{\text{sg}}$ does not correspond to following the gradients of any one scalar objective $J$ (section B.1). This is because the optimization dynamics are in general *non-conservative*. The stopgrad structure breaks the symmetry required for the updates to be the gradient of a single scalar function. In this sense, SGFlow is formally a (two-player) *game* rather than standard gradient descent on one potential function, albeit a trivial one where the main player controls all parameters except those in the stopgrad, and the stopgrad player keeps a virtual copy of the parameters that simply equal the first players parameters. Put another way, the optimization dynamics if taken in the limit of small step size correspond to non-conservative / non-gradient vector flow.

## 4 EXPERIMENTS

### 4.1 IMAGE MODELING ON CIFAR-10

**Architecture.** We modify the existing diffusion U-Net from Dhariwal and Nichol (2021) by embedding both $t$ and $u$ with the usual Fourier embeddings and then concatenate on input to a small MLP that maps the two times to a hidden representation for use in the network. We use 128 channels and channel multipliers set to (1,2,2,2) with `attention` set to (False, False, True, False).

**Training Settings.** We use dropout 0.1. We do not condition on the class label We use $\alpha_t = 1 - t$ and $\sigma_t = t$, with noise at $X_0$ and data at $X_1$. We train for 200,000 steps at learning rate 2e-4.

**Losses.** We benchmark based on Flow Matching (Lipman et al., 2022) as the reference method for sampling in the $O(100)$ step regime. For few-step sampling, we benchmark with Meanflow (Geng et al., 2025), which `StopGrad`'s all model derivatives in the loss to avoid backpropagation through differentiation. We also benchmark other recently proposed methods from Boffi et al. (2024): the Lagrangian loss, Eulerian loss, and Progressive loss (discussed in Section 5).

| Method | 10 steps | 50 steps | 100 steps | theory |
|---|---|---|---|---|
| Flow Matching | 24.87 | 3.53 | 3.05 | yes |
| Lagrange | 248.76 | 230.43 | 221.22 | yes |
| Euler | 77.19 | 66.99 | 38.95 | yes |
| Progressive | 337.36 | 235.20 | 206.18 | yes |
| Meanflow | 37.32 | 4.54 | 4.23 | no |
| SGFlow | **12.26** | **2.88** | **2.81** | **yes** |

Table 2: FID scores versus sampling steps on CIFAR-10 computed from 50,000 EMA samples after 200,000 training steps. The "theory" column means whether the stationary points of the optimization have been proven to exist if and only if the function is the flow map that integrates the ODE.

**Results.** We report the Frechet Inception Distance (FID) (Heusel et al., 2017) in Table 2. We find that SGFlow produces better FID than MeanFlow at each choice of sampling steps for the given (rather common for CIFAR) training configuration.

### 4.2 MEMORY USAGE

In the last column of Table 1, we note whether a method requires differentiation through an iterated model call. As discussed in Section 5, for a flow map model $f_\theta = x + (u - t)\tilde{f}_\theta$, the Lagrangian loss involves a nested evaluation $\tilde{f}_\theta(u, u, f_\theta(t, u, x))$, with a similar nesting for the Progressive loss. The Eulerian loss requires computing $\nabla_x f_\theta(t, u, x)\, \tilde{f}_\theta(t, t, X_t)$, which entails a product-rule expansion. Here we empirically compare the *peak GPU memory usage during the backward pass* for different losses, holding the architecture and data size fixed (the U-Net with a batch of CIFAR images).

**Results.** We demonstrate the peak GPU memory usage during backward pass in Table 3. As expected, the Lagrange, Euler, and Progressive losses are the most memory-intensive, reflecting the need to backpropagate through nested model evaluations or product-rule terms. On the other end, MeanFlow, which computes Jacobian–vector products but detaches the full JVP, has the lowest memory usage, though the effect of this detachment on optimization dynamics is not analyzed in that work. SGFlow falls between these extremes, striking a balance between memory usage, preserving the optimum, and benefiting from optimizing through the model derivatives in the loss.

| Flow Matching | MeanFlow | SGFlow | Lagrange | Euler | Progressive |
|---|---|---|---|---|---|
| 16.8 Gb | 14.2 Gb | 43.2 Gb | 69.8 Gb | 69.8b G | 54.3 Gb |

Table 3: **Peak GPU memory usage during backward pass (in Gb).** Values reflect the maximum allocated memory measured across the training step's backward pass (i.e., during gradient computation). SGFlow strikes a balance in GPU memory usage, preserves the true flow map as an optimum, and optimizes through all model derivatives without detaching.

## 5 RELATED WORK

Sampling from continuous-time generative models such as diffusion and flow models requires numerical integration. Each integration step requires a forward pass of a neural network, leading to computational costs and slow sampling. Current approaches to address this cost can be broadly categorized into two types (1) distilling a pretrained diffusion or flow model into a few-step solver (Salimans and Ho, 2022; Kim et al., 2023; Liu et al., 2023), and (2) learning a few-step solver (Zhou et al., 2025). Some approaches in this area allow for distillation as well as training from scratch (Song et al., 2023; Boffi et al., 2024; Boffi and Vanden-Eijnden, 2023).

Consistency Models (CMs) (Song et al., 2023; Song and Dhariwal, 2023; Lu and Song, 2024) learn a one-step map from noise to data, either by distilling a pretrained model or by learning from scratch. Distillation requires sampling trajectories from the teacher model. To allow for more steps after either training approach, CMs iteratively re-noise the one-step solution back to successively smaller time under the interpolant and then denoise, but this can take the solver off the probability flow.

Consistency trajectory models (CTMs) (Kim et al., 2023) extend CMs to learn two-time maps using a combination of consistency and adversarial objectives, which requires training an additional discriminator model (Goodfellow et al., 2014). CTM and Gameflow both target the same mathematical object, the probability–flow ODE flow map (i.e., the integral of the ODE), but they learn this map through different means. CTM learns the map by distilling a teacher solver, and the losses for teacher and student involve several nested model evaluations (with data at $x_0$, for $0 \leq s \leq u \leq t \leq 1$, the teacher integrates from $t$ to $u$, then jumps from $u$ to $s$, then from $s$ to 0; and the student jumps from $t$ to $s$ and then to 0). The objective depends on a chosen feature-space distance and, in practice, includes DSM and GAN terms that further influence the optimum. Consequently, the CTM loss is sensitive to the quality of the ODE discretization used by the teacher (in practice CTM finds the need to use a 2nd order solver during training) and necessitates the presence of the GAN.

Inductive Moment Matching (IMM) (Zhou et al., 2025) learns a few-step model via an implicit generative model trained with MMD (Smola et al., 2006; Gretton et al., 2012), with the MMD which is estimated biasedly within subsets of data. In practice, the authors must use time-weighting schedules and specific curriculum/inductive procedure to stabilize optimization. While IMM produces high-quality image samples, it solves the problem of marginally sampling the data distribution rather than sampling along a probability flow, where the latter is the task studied in this work.

MeanFlow (Geng et al., 2025) derives a JVP-based objective for flow maps. They train $\tilde{f}$ to bring $x_u$ down to $X_t$ via the parameterization $f_\theta = x + (u - t)\tilde{f}_\theta$, and train $\tilde{f}$ via the following loss:

$$\mathcal{L}_{t,u}^{\text{meanflow}} := \mathbb{E}[\|\tilde{f}_\theta(t, u, X_t) - \dot{x}_u + (u - t)(\text{sg}[\partial_x \tilde{f}_\theta]_{(t,u,X_t)} \cdot \dot{x}_u + \text{sg}[\partial_u \tilde{f}_\theta]_{(t,u,X_t)})\|^2]. \quad (11)$$

Applying the stopgrad sg to all model derivatives improves efficiency, but there are no differentiated loss terms that encourage the model derivatives $\partial_t \hat{f}$ and $\partial_x \hat{f}$ to move toward the true flow map derivatives. This contrasts SGFlow where $\text{sg}[\tilde{f}_\theta(t, t, x)]$ is used in place of $\mathbb{E}[\dot{X}_t | X_t]$, but where another term in the loss trains these two quantities to match. Finally, between equations (10, 11) in Geng et al. (2025), $v$ is replaced with $\dot{X}_t$ where it appears quadratically, thereby pulling an expectation through a square and missing a resulting trace covariance term (Boffi et al., 2025).

Flow Map Matching (Boffi et al., 2024) learns a two-time flow map. This allows for mapping along the probability flow in either direction, without adversarial training. Their Lagrangian Flow Map Matching loss requires only time derivatives, but an additional invertibility loss encouraging invertibility via time-swapping so that $\hat{f}(t, u, \hat{f}(u, t, x)) \approx x$. While straightforward to compute, gradient steps require evaluating the model and its inverse at each step of training.

Boffi et al. (2025) optimize velocity matching along with one of the three flow map functionals. For the parameterization $f_\theta := x + (u - t)\tilde{f}_\theta$, the first one is:

$$\text{LSD} : \mathcal{L}_{t,u}^{\text{lagrange}} := \mathbb{E}[\|\partial_u f_\theta(t, u, X_t) - \tilde{f}_\theta(u, u, f_\theta(t, u, X_t))\|^2]$$

LSD comes from the condition $\partial_t f(t, u, x) = v(u, x_u) = v(u, f(t, u, X_t))$ and uses $\partial_t f(t, t, \cdot) = -v(t, \cdot) \implies \tilde{f}(t, t, \cdot) = v(t, \cdot)$. It is a variant of the Lagrangian loss from Boffi et al. (2024). The next is:

$$\text{ESD} : \mathcal{L}_{t,u}^{\text{euler}} := \mathbb{E}[\|\partial_t f_\theta(t, u, X_t) + \nabla_x f_\theta(t, u, X_t)\tilde{f}_\theta(t, t, X_t)\|^2].$$

ESD comes from condition $\partial_t f(t, u, x) + \nabla_x f(t, t, x)v(t, x) = 0$ where in the loss, $v$ is replaced with $\tilde{f}(t, t, x)$. The last one, for an intermediate time $m$, is:

$$\text{PSD} : \mathcal{L}_{t,u}^{\text{progress}} := \mathbb{E}[\|f_\theta(t, u, X_t) - f_\theta(m, u, f_\theta(t, m, X_t))\|^2]$$

PSD comes from the composition property: $f(t, u, x) = f(m, u, f(t, m, x))$ for $m \in (t, u)$. SGFlow, LSD, ESD, and PSD all aim to enforce a similar set of flow map properties. $\mathcal{L}^{\text{lagrange}}$ and $\mathcal{L}^{\text{progress}}$ must optimize through nested model evaluations, doubling the computational graph for backprop. $\mathcal{L}^{\text{euler}}$ optimizes through $\nabla_x f_\theta \cdot \tilde{f}_\theta$, which causes reverse-mode autodifferentiation to invoke the full product rule term, $\nabla_\theta[\nabla_x f_\theta \cdot \tilde{f}_\theta] = [\nabla_\theta \nabla_x f_\theta] \cdot \tilde{f}_\theta + [\nabla_x f_\theta] \cdot \nabla_\theta \tilde{f}_\theta$. To alleviate these challenges and improve optimization, Boffi et al. (2025) suggest specific `StopGrad` options for LSD, ESD, and PSD to improve optimization. It would be interesting to more deeply understand why these stopgrads help given their empirical success for flow map losses.

*Distillation methods.* A complementary line of work approaches flow map learning by *distilling* the outputs of pretrained flow matching velocity models into few-step solvers. Specifically the unknown $v()$ in the flow map identities is taken to be a pretrained network. By contrast, we emphasize training from scratch, avoiding dependence on a teacher model and ensuring that all components of the flow map are learned end-to-end. That said, distillation can be attractive in practice when a pretrained model is already trusted (when $v_\theta$ corresponds to the endpoint distributions and the chosen $\alpha_t$ and $\sigma_t$, or when the objective is weaker—for example, to marginally sample from the approximated data distribution without explicitly solving the probability flow ODE).

## 6 DISCUSSION AND LIMITATIONS

**Identifying functions through PDEs**  *Given a PDE solved by a sought-after mapping $f$, featuring a combination of terms such as $\partial_t f, \partial_x f, v$ that should be set to 0, which terms should be parameterized by the model and which should be approximated as part of a ground-truth loss target?* If the equations can be rewritten in several ways, which yield easier or more challenging objectives? Answering this is applicable to improving training objectives for generative models as well as more generally solving PDEs with machine learning.

**Invertibility**  The loss targets an invertible function at optimum. To simplify training, we explicitly give up knowing the inverse, meaning that we only learn maps in one direction. Luckily, this is the usual scenario for generative modeling. For likelihoods, one can still substitute $-\partial_t \hat{f}$ for $v_\theta$ in the probability flow ODE (Song et al., 2021; Boffi and Vanden-Eijnden, 2023). Thus this method can be seen from the perspective of training a normalizing flow (Tabak and Vanden-Eijnden, 2010; Tabak and Turner, 2013; Rezende and Mohamed, 2015; Papamakarios et al., 2021) without requiring the invertible architecture or inverse-dependent loss.

**Architectures.**  SGFlow, Flow Map Matching, Simplified Consistency Models, and MeanFlow all specify models whose *time-derivative* equal the target of diffusion model training, but directly adapt architectures meant for diffusion models themselves. Example architectures used in these works are the UNet from Dhariwal and Nichol (2021), the diffusion transformer from Peebles and Xie (2023); Ma et al. (2024), and the EDM architecture from Karras et al. (2022; 2024). These architectures may thus be suboptimal for the problem at hand, precisely because the target of interest is defined as a function often computed in many diffusion model forward passes (an integral). In this work, compute limitations did not allow for the thorough exploration of architectures, but the authors believe that rethinking architectures is a convincing direction to improve the quality and training-efficiency of learned flow maps.

ACKNOWLEDGMENTS

This work was partly supported by the NIH/NHLBI Award R01HL148248, NSF Award 1922658 NRT-HDR: FUTURE Foundations, Translation, and Responsibility for Data Science, NSF CAREER Award 2145542, NSF Award 2404476, ONR N00014-23-1-2634, Optum, and Apple. This work was also supported by IITP with a grant funded by the MSIT of the Republic of Korea in connection with the Global AI Frontier Lab International Collaborative Research. Mark Goldstein would like to thank the Simons Foundation and Flatiron Institute for funding and compute resources that supported this project. Anshuk Uppal would like to thank the Centre for Basic Machine Learning Research in Life Sciences, and Technical University Denmark.

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

# A PROOFS FOR STATIONARY POINTS

## A.1 FIRST VARIATION DEFINITIONS

We consider scalar-valued loss functions $\mathcal{L} : \mathcal{F} \to \mathbb{R}$ that map a function $f \in \mathcal{F}$ to a real value.

Define the tangent space $\mathcal{T}_f(\mathcal{F})$ at $f$. This space contains functions $h \in \mathcal{T}_f(\mathcal{F})$ such that there exists a curve indexed by scalar $\epsilon$ such that for each $\epsilon$, $f_\epsilon \in \mathcal{F}$, and we have that $f_0 = f$ and $(\frac{d}{d\epsilon} f_\epsilon)|_{\epsilon=0} = h$.

The first variation $\delta\mathcal{L}$ of such a functional $\mathcal{L}$ evaluated at $f \in \mathcal{F}$ in direction $h \in \mathcal{T}_f(\mathcal{F})$ is defined as:

$$\delta\mathcal{L}[f; h] := \left( \frac{d}{d\epsilon} \mathcal{L}[f + \epsilon h] \right)_{\epsilon=0} \tag{12}$$

We then have that $f^*$ is a stationary point w.r.t. $\mathcal{F}$ if $\delta[f; h] = 0$ for all $h \in \mathcal{T}_f(\mathcal{F})$.

## A.2 STOPGRAD FOR FUNCTIONALS

We define the stopgrad symbol sg for a functional as follows. Let $\mathcal{O}$ be a functional that maps two functions $f, g$ to a real value. Let $\mathcal{L}[f]$ be a functional that is written in terms of $\mathcal{O}$ with symbol sg as $\mathcal{L}[f] := \mathcal{O}[f, \mathrm{sg}[f]]$, then we evaluate the following two quantities as follows

$$\mathcal{L}[f] = \mathcal{O}[f, f] \tag{13}$$
$$\delta\mathcal{L}[f; h] = \delta\mathcal{O}[f, f; h, 0] \tag{14}$$

That is, the functional evaluates as usual but in a first variation, we do not perturb terms in sg. This corresponds to the stopgrad or detach() operation used in machine learning code with autodifferentiation.

### A.3 First Variation of Original Loss Functional

Our functional $\mathcal{L}[\tilde{f}]$ acts on functions $\tilde{f}$. According to the definitions in Section A.1, we need to compute $\delta\mathcal{L}[\tilde{f}; h] = (\frac{d}{d\epsilon}\mathcal{L}[\tilde{f} + \epsilon h])_{\epsilon=0}$. The functional is:

$$T_1[\tilde{f}] := \|\|(\partial_t f)_{(t,u,x_t)} + (\partial_x f)_{(t,u,x_t)}\dot{x}_t\|_{f=x+(u-t)\tilde{f}}^2$$

$$T_2[\tilde{f}] := \|(\partial_x f)_{(t,u,x_t)}(\dot{x}_t - \mathbb{E}[\dot{x}_t|x_t])\|_{f=x+(u-t)\tilde{f}}^2$$

$$A[\tilde{f}] = T_1[\tilde{f}] - T_2[\tilde{f}]$$

$$\mathcal{L}[\tilde{f}] = \mathbb{E}_{q(t,u),q(x_0),q(x_1)}\Big[A[\tilde{f}]\Big]$$

Lets define the path $f_\epsilon$ by replacing $\tilde{f}$ with $\tilde{f}_\epsilon := \tilde{f} + \epsilon h$. Then:

$$f_\epsilon := x + (u-t)\tilde{f}_\epsilon = x + (u-t)(\tilde{f} + \epsilon h) = x + (u-t)\tilde{f} + \epsilon(u-t)h \tag{15}$$

Then

$$\frac{d}{d\epsilon}\mathcal{L}[\tilde{f} + \epsilon h] = \mathbb{E}\Big[\frac{d}{d\epsilon}A[\tilde{f}_\epsilon]\Big] = \mathbb{E}\Big[\frac{d}{d\epsilon}T_1[\tilde{f} + \epsilon h] - \frac{d}{d\epsilon}T_2[\tilde{f} + \epsilon h]\Big] \tag{16}$$

We first compute this derivative and then evaluate it at $\epsilon = 0$.

So

$$\partial_t f_\epsilon = \partial_t\Big[x + (u-t)\tilde{f} + \epsilon(u-t)h\Big] \tag{17}$$

$$= \partial_t x + \partial_t\Big[(u-t)\tilde{f}\Big] + \epsilon\partial_t\Big[(u-t)h\Big] \tag{18}$$

$$= (u-t)\partial_t\tilde{f} - \tilde{f} + \epsilon\Big((u-t)\partial_t h - h\Big) \tag{19}$$

$$= (u-t)(\partial_t\tilde{f} + \epsilon\partial_t h) - (\tilde{f} + \epsilon h) \tag{20}$$

and

$$\partial_x f_\epsilon = \partial_x\Big[x + (u-t)\tilde{f} + \epsilon(u-t)h\Big] = I + (u-t)\partial_x(\tilde{f} + \epsilon h) \tag{21}$$

and

$$\frac{d}{d\epsilon}\partial_t f_\epsilon = \frac{d}{d\epsilon}\Big[(u-t)(\partial_t\tilde{f} + \epsilon\partial_t h) - (\tilde{f} + \epsilon h)\Big] \tag{22}$$

$$= \frac{d}{d\epsilon}\Big[(u-t)\partial_t\tilde{f} + \epsilon(u-t)\partial_t h - \tilde{f} - \epsilon h\Big] \tag{23}$$

$$= \frac{d}{d\epsilon}\Big[\epsilon(u-t)\partial_t h - \epsilon h\Big] \tag{24}$$

$$= (u-t)\partial_t h - h \tag{25}$$

and

$$\frac{d}{d\epsilon}\partial_x f_\epsilon = \frac{d}{d\epsilon}\Big[I + (u-t)\partial_x(\tilde{f} + \epsilon h)\Big] \tag{26}$$

$$= \frac{d}{d\epsilon}I + \frac{d}{d\epsilon}(u-t)\partial_x\tilde{f} + \frac{d}{d\epsilon}(u-t)\partial_x\epsilon h \tag{27}$$

$$= (u-t)\partial_x h \tag{28}$$

For the first term,

$$T_1[\tilde{f} + \epsilon h] = \|\partial_t f_\epsilon + (\partial_x f_\epsilon)\dot{x}_t\|^2$$

Differentiating

$$\frac{d}{d\epsilon}T_1[\tilde{f} + \epsilon h] = 2\Big(\partial_t f_\epsilon + (\partial_x f_\epsilon)\dot{x}_t\Big)^\top\frac{d}{d\epsilon}\Big(\partial_t f_\epsilon + (\partial_x f_\epsilon)\dot{x}_t\Big) \tag{29}$$

$$= 2\Big(\partial_t f_\epsilon + (\partial_x f_\epsilon)\dot{x}_t\Big)^\top\Big(\underbrace{(u-t)\partial_t h - h} + \underbrace{(u-t)\partial_x h\,\dot{x}_t}\Big) \tag{30}$$

$$= 2\Big( \underbrace{(u-t)(\partial_t \tilde{f} + \epsilon \partial_t h) - (\tilde{f} + \epsilon h)} + \underbrace{(I + (u-t)\partial_x(\tilde{f} + \epsilon h))}\dot{x}_t \Big)^\top \tag{31}$$

$$\Big( \underbrace{(u-t)\partial_t h - h} + \underbrace{(u-t)\partial_x h}\, \dot{x}_t \Big) \tag{32}$$

So

$$\frac{d}{d\epsilon} T_1[\tilde{f} + \epsilon h]\Big|_{\epsilon=0} = 2\Big((u-t)(\partial_t \tilde{f} + \partial_x \tilde{f}\dot{x}_t) - \tilde{f} + \dot{x}_t\Big)^\top \Big((u-t)(\partial_t h + \partial_x h \dot{x}_t) - h\Big) \tag{33}$$

For the second term,

$$T_2[\tilde{f} + \epsilon h] = \|\partial_x f_\epsilon(\dot{x}_t - v)\|^2 \tag{34}$$

and

$$\frac{d}{d\epsilon} T_2[\tilde{f} + \epsilon h] = 2\Big[\partial_x f_\epsilon(\dot{x}_t - v)\Big]^\top \frac{d}{d\epsilon}\Big[\partial_x f_\epsilon(\dot{x}_t - v)\Big] \tag{35}$$

$$= 2\Big[\partial_x f_\epsilon(\dot{x}_t - v)\Big]^\top \frac{d}{d\epsilon}(\partial_x f_\epsilon)(\dot{x}_t - v) \tag{36}$$

$$= 2\Big[\Big(I + (u-t)\partial_x(\tilde{f} + \epsilon h)\Big)(\dot{x}_t - v)\Big]^\top (u-t)\partial_x h(\dot{x}_t - v) \tag{37}$$

So

$$\frac{d}{d\epsilon} T_2[\tilde{f} + \epsilon h]\Big|_{\epsilon=0} = 2\Big[\Big(I + (u-t)\partial_x \tilde{f}\Big)(\dot{x}_t - v)\Big]^\top (u-t)\partial_x h(\dot{x}_t - v) \tag{38}$$

So combining

$$\frac{d}{d\epsilon} A\Big|_{\epsilon=0} = 2\Big((u-t)(\partial_t \tilde{f} + \partial_x \tilde{f}\dot{x}_t) - \tilde{f} + \dot{x}_t\Big)^\top \Big((u-t)(\partial_t h + \partial_x h \dot{x}_t) - h\Big) \tag{39}$$

$$- 2\Big[\Big(I + (u-t)\partial_x \tilde{f}\Big)(\dot{x}_t - v)\Big]^\top (u-t)\partial_x h(\dot{x}_t - v) \tag{40}$$

At $t = u$, this simplifies

$$1[t = u]\frac{d}{d\epsilon} A[\tilde{f}_\epsilon]\Big|_{\epsilon=0} = 2(\dot{x}_t - \tilde{f})^\top(-h) \tag{41}$$

which is the first variation for regression that makes $\tilde{f}$ equal to $E[\dot{x}_t | x_t]$. Summarizing,

$$\delta\mathcal{L}[\tilde{f}; h] = \mathbb{E}\Big[2\Big((u-t)(\partial_t \tilde{f} + \partial_x \tilde{f}\dot{x}_t) - \tilde{f} + \dot{x}_t\Big)^\top \Big((u-t)(\partial_t h + \partial_x h \dot{x}_t) - h\Big) \tag{42}$$

$$- 2\big[(I + (u-t)\partial_x \tilde{f})(\dot{x}_t - v)\big]^\top (u-t)\partial_x h(\dot{x}_t - v)\Big] \tag{43}$$

A.4  LEMMA: VELOCITY MATCHES AT A STATIONARY POINT OF ORIGINAL FUNCTIONAL

**Lemma 1.** *Let $\tilde{f}^*$ be a stationary point of $\mathcal{L}$. Assume that $\tilde{f}^*$ is bounded. Assume that $\tilde{f}^*, v \in C^1$ in arguments $(t, u, x)$ and that all expectations of terms featured in the integrand of $\mathcal{L}$ (i.e., $v, \tilde{f}, \partial_t \tilde{f}, \partial_u \tilde{f}, \partial_x \tilde{f}, \ldots$) are finite. Then we have that $\tilde{f}^*(t, t, \cdot) = v(t, \cdot)$ where the velocity $v(t, x) = \mathbb{E}[\dot{x}_t | x_t]$.*

*Proof.* We proceed by contradiction. By the premise, we are at a stationary point $\tilde{f}^*$. Let $f^* := x + (u - t)\tilde{f}^*$. By the definition of stationary point in section A.1, we have that $\delta\mathcal{L}[\tilde{f}^*; h] = 0$ for all admissible $h$. Suppose for the sake of contradiction that at this stationary point, the velocity does not match, meaning

$$-\partial_t f^*(t, t, \cdot) = \tilde{f}^*(t, t, \cdot) \underbrace{\neq}_{\text{suppose for contradiction}} v(t, \cdot) \tag{44}$$

The proof proceeds by picking a direction for which the first variation is nonzero, providing a contradiction to being at a stationary point. The contradiction (the direction for which the first variation is nonzero) is constructed to arise from assuming that the velocity does not match, meaning that by contradiction the velocity does not match. Specific care is taken to ensure that this direction is admissible, in this case meaning it is a continuous function.

We name a sequence of functions $g_\eta$ such there exists $\eta^*$ such that $g_{\eta^*}$ is continuous but yields the nonzero variation when chosen as a direction. To establish this is existence under continuity, the dominated convergence theorem is used.

Let us define $g(t, u, x) = 1[t = u]\left(\tilde{f}^*(t, u, x) - v(t, x)\right)$ and evaluate it at $t = u$ so that $g(t, t, x) = \tilde{f}^*(t, t, x) - v(t, x)$. We then define the soft indicator $I_\eta(t, u)$ that goes to $1[t = u]$ as $\eta \to 0$ and define it as:

$$I_\eta(t, u) = 1[\eta > 0]2\left(1 - \frac{1}{1 + \exp(-\frac{1}{\eta^2}(t - u)^2)}\right) + 1[\eta = 0]1[t = u] \tag{45}$$

Using the soft indicator, we define a sequence of functions $g_\eta$ so that as $\eta \to 0$ we will have pointwise convergence of $g_\eta(t, t, x) \to g(t, t, x)$ which also means that $g_\eta(t, u, x)$ for $t \neq u$ goes to 0. We pick

$$g_\eta(t, u, x) = I_\eta(t, u)\left(\tilde{f}^*(t, t, x) - v(t, x)\right) \tag{46}$$

**Pointwise convergence of g in eta.** We first establish pointwise convergence of $g_\eta$ to $g$ for all arguments $(t, u, x)$ as $\eta \to 0$ from the right.

$$\forall \hat{\eta} \geq 0, \quad \lim_{\eta \to (\hat{\eta})^+} g_\eta(t, u, x) = g_{\hat{\eta}}(t, u, x) \tag{47}$$

To show this, for any $\hat{\eta} > 0$, use continuity of $g_{\hat{\eta}}(t, u, x)$ in $\hat{\eta}$ (product of function without $\eta$ times the soft indicator which is continuous). Then to establish for $\hat{\eta} = 0$, we consider two cases $t = u$ and $t \neq u$. For equality:

$$\lim_{\eta \to 0^+} g_\eta(t, t, x) \tag{48}$$

$$= \lim_{\eta \to 0^+} I_\eta(t, t)\left(\tilde{f}^*(t, t, x) - v(t, x)\right) \tag{49}$$

$$= \lim_{\eta \to 0^+}\left[1[\eta > 0]2\left(1 - \frac{1}{1 + \exp(0)}\right) + 1[\eta = 0]\right]\left(\tilde{f}^*(t, t, x) - v(t, x)\right) \tag{50}$$

$$= \lim_{\eta \to 0^+}\left[1[\eta > 0]1 + 1[\eta = 0]\right]\left(\tilde{f}^*(t, t, x) - v(t, x)\right) \tag{51}$$

$$= \lim_{\eta \to 0^+} 1[\eta \geq 0]\left(\tilde{f}^*(t, t, x) - v(t, x)\right) \tag{52}$$

$$= \tilde{f}^* - v \tag{53}$$

which equals $g_{\eta=0}(t,t,x)$, establishing continuity. Now for $t \neq u$. $\forall \delta > 0$ we must name an $\eta(\delta)$ such that $g_{\eta(\delta)}$ such that $|g_{\eta(\delta)} - g_0| < \delta$ i.e. $|g_{\eta(\delta)} - 0| < \delta$. Assume $|\tilde{f}^* - v| < k$ uniformly in all input values t,x.

$$\lim_{\eta \to 0^+} g_\eta(t, u, x)$$

$$= \lim_{\eta \to 0^+} \left[ 1[\eta > 0]2\left(1 - \frac{1}{1 + \exp(-\frac{1}{\eta^2}(t-u)^2)}\right) + 1[\eta = 0]1[t = u] \right] \left( \tilde{f}^*(t, t, x) - v(t, x) \right)$$

$$= \lim_{\eta \to 0^+} \left[ 1[\eta > 0]2\left(1 - \frac{1}{1 + \exp(-\frac{1}{\eta^2}(t-u)^2)}\right) \right] \left( \tilde{f}^*(t, t, x) - v(t, x) \right)$$

Since we are finding a $\delta$ and $\eta(\delta)$ so that $|g_{\eta(\delta)} - 0| < \delta$ which means $|g_{\eta(\delta)}| < \delta$, this just means we can set $\delta$ to an upper bound on the term we are limiting: let the indicator take on 1 as when it is 0 we are done.

$$\delta = \left| 2(1 - \frac{1}{1 + \exp(...)}) \right| k \tag{54}$$

$$\iff \frac{\delta}{k} = 2(1 - \frac{1}{1 + \exp(-\frac{1}{\eta^2}(t-u)^2)}) \tag{55}$$

$$\iff \frac{\delta}{2k} = 1 - \frac{1}{1 + \exp(-\frac{1}{\eta^2}(t-u)^2)} \tag{56}$$

$$\iff 1 - \frac{\delta}{2k} = \frac{1}{1 + \exp(-\frac{1}{\eta^2}(t-u)^2)} \tag{57}$$

$$\iff \frac{2k}{2k} - \frac{\delta}{2k} = \frac{1}{1 + \exp(-\frac{1}{\eta^2}(t-u)^2)} \tag{58}$$

$$\iff \frac{2k - \delta}{2k} = \frac{1}{1 + \exp(-\frac{1}{\eta^2}(t-u)^2)} \tag{59}$$

$$\iff \frac{2k}{2k - \delta} = 1 + \exp(-\frac{1}{\eta^2}(t-u)^2) \tag{60}$$

$$\iff \frac{2k}{2k - \delta} - 1 = \exp(-\frac{1}{\eta^2}(t-u)^2) \tag{61}$$

$$\iff \frac{2k}{2k - \delta} - \frac{2k - \delta}{2k - \delta} = \exp(-\frac{1}{\eta^2}(t-u)^2) \tag{62}$$

$$\iff \frac{\delta}{2k - \delta} = \exp(-\frac{1}{\eta^2}(t-u)^2) \tag{63}$$

$$\iff \log\frac{\delta}{2k - \delta} = -\frac{1}{\eta^2}(t-u)^2 \tag{64}$$

$$\iff -\log\frac{\delta}{2k - \delta} = \frac{1}{\eta^2}(t-u)^2 \tag{65}$$

$$\iff -\frac{\log\frac{\delta}{2k-\delta}}{(t-u)^2} = \frac{1}{\eta^2} \tag{66}$$

$$\iff -\frac{(t-u)^2}{\log\frac{\delta}{2k-\delta}} = \eta^2 \tag{67}$$

$$\tag{68}$$

Now note that the soft indicator is strictly $< 1$ and that $g_\eta$ for fixed $(t, u, x)$ is between $-k$ and 0 or 0 and $k$ depending on the sign of $\tilde{f}^* - v$, but never both. So its magnitude is at most $k$. So

$$|g_{\eta(\delta)} - g_0| = |g_{\eta(\delta)} - 0| < \delta < k \tag{69}$$

This can help us ascertain that the above square root to solve for $\eta$ will be well defined:

$$\delta < k \implies 2k - \delta > k \tag{70}$$

$$\implies \frac{1}{2k-\delta} < \frac{1}{k} \tag{71}$$

$$\implies \frac{\delta}{2k-\delta} < \frac{\delta}{k} \tag{72}$$

$$\implies \frac{\delta}{2k-\delta} < 1 \tag{73}$$

$$\implies \log \frac{\delta}{2k-\delta} < \log 1 \tag{74}$$

$$\implies \log \frac{\delta}{2k-\delta} < 0 \tag{75}$$

meaning

$$\eta(\delta) = \sqrt{\frac{(t-u)^2}{|\log \frac{\delta}{2k-\delta}|}} \tag{76}$$

thus establishing convergence of $g_\eta \to g$ as $\eta \to 0^+$ for each $(t, u, x)$ i.e. pointwise convergence.

Now recall the fist variation of $\mathcal{L}$ (section A.3) and consider it as a function of $\eta$:

$$s(\eta) := \delta\mathcal{L}[\tilde{f}^*; g_\eta] = \mathbb{E}\Bigg[ 2\Big((u-t)(\partial_t \tilde{f}^*) - \tilde{f}^* + (I + (u-t)(\partial_x \tilde{f}^*))\dot{x}_t\Big)^\top \tag{77}$$

$$\Big((u-t)\partial_t g_\eta - g_\eta + (u-t)(\partial_x g_\eta)\dot{x}_t\Big) \tag{78}$$

$$- 2\Big[\Big(I + (u-t)\partial_x \tilde{f}^*\Big)(\dot{x}_t - v)\Big]^\top (u-t)\partial_x g_\eta(\dot{x}_t - v)\Bigg] \tag{79}$$

**Pointwise convergence of integrand in eta.** Collect the variables $\omega = (t, u, x_0, x_1)$ and recall that $x_t$ and $\dot{x}_t$ are functions of $(x_0, x_1)$. Define $\phi_\eta(\omega)$ as shorthand for the expectand so that $s(\eta) = \mathbb{E}[\phi_\eta(\omega)]$. Under the boundedness assumptions and noting that $\phi_\eta$ only polynomially combines $g_\eta$ with $(\partial_t \tilde{f}^*, \partial_u \tilde{f}^*, \partial_x \tilde{f}^*, v, \partial_x v, \dots)$, similar reasoning used to show $g_\eta \to g$ can also be used to establish that $\phi_\eta \to \phi$ pointwise.

**Establish upper envelope.** In addition to pointwise convergence of $\phi_\eta \to \eta$ as $\eta \to 0$ from the right, we need an upper envelope $G(\omega)$. Beyond the assumptions, the only thing needed to show that an upper envelope exists is to control the term $|(u-t)\partial_t I_\eta(t, u)|$. The derivative of the soft indicator appears since $\partial_t g_\eta = \partial_t(I_\eta g) = (\partial_t I_\eta)g + I_\eta \partial_t g$. At $\eta = 0$ we have $I_\eta(t, u) = 1[t = u]$, so $(u-t)\partial_t I_\eta(t, u)$ vanishes identically: it is 0 for $t \neq u$ because $I_0$ is constant, and for $t = u$ because of the $(u-t)$ prefactor. For $\eta > 0$, define:

$$z := \frac{(t-u)^2}{\eta^2}, \quad \sigma(z) := \frac{1}{1 + \exp(-z)}, \quad \sigma'(z) := \frac{\exp(-z)}{[1 + \exp(-z)]^2} \tag{80}$$

Note that for $\eta > 0$,

$$\partial_t I_\eta(t, u) = 2\frac{2(t-u)}{\eta^2}\sigma'(\frac{(t-u)^2}{\eta^2}) \tag{81}$$

and so

$$|(u-t)\partial_t I_\eta(t, u)| = 2(u-t)\frac{2(u-t)}{\eta^2}\sigma'(\frac{(t-u)^2}{\eta^2}) = 4z\sigma'(z) \leq \sup_{z \geq 0} 4z\sigma'(z) \leq C_0 < \infty \tag{82}$$

Because $r(z) := 4z\sigma'(z)$ is continuous and satisfies $r(0) = 0$ and $r(z) \to 0$ as $z \to \infty$, it attains a finite maximum. **This bound is independent of $\eta$.** Thus every term containing $(u-t)\partial_t g_\eta$ is uniformly bounded in $\eta$ by a product of a constant (from the bound and $I_\eta \in [0, 1)$). The other quantities in $\phi$ are bounded by assumption. Thus such a bounding envelope $G(\omega)$ exists.

**Using dominated convergence.** First,

$$\lim_{\eta \to 0^+} s(\eta) = \lim_{\eta \to 0^+} \mathbb{E}[\phi_\eta] = \lim_{\eta \to 0^+} p(t = u)\mathbb{E}[\phi_\eta \mid t = u] + p(t < u)\mathbb{E}[\phi_\eta \mid t < u] \tag{83}$$

By the pointwise convergence of $\phi_\eta \to \phi$ as $\eta \to 0^+$ and by the envelope, we can compute the limit of the first and second terms separately. Expanding the first term (with $p = u$):

$$\lim_{\eta \to 0^+} p(t = u)\mathbb{E}[\phi_\eta \,|\, t = u] = \lim_{\eta \to 0^+} p(t = u)\mathbb{E}\Bigg[ 2\Big((u - t)(\partial_t \tilde{f}^*) - \tilde{f}^* + (I + (u - t)(\partial_x \tilde{f}^*))\dot{x}_t\Big)^\top$$

$$\Big((u - t)\partial_t g_\eta - g_\eta + (u - t)(\partial_x g_\eta)\dot{x}_t\Big)$$

$$- 2\Big[\Big(I + (u - t)\partial_x \tilde{f}^*\Big)(\dot{x}_t - v)\Big]^\top (u - t)\partial_x g_\eta(\dot{x}_t - v) \,|\, t = u\Bigg]$$

$$= \lim_{\eta \to 0^+} p(t = u)\mathbb{E}\Bigg[ 2\Big(-\tilde{f}^* + \dot{x}_t\Big)^\top \Big(-g_\eta\Big) \,|\, t = u\Bigg]$$

$$= \lim_{\eta \to 0^+} p(t = u)\mathbb{E}\Bigg[ 2\Big(-\tilde{f}^* + \mathbb{E}[\dot{x}_t \,|\, x_t]\Big)^\top \Big(-g_\eta\Big) \,|\, t = u\Bigg]$$

$$= \lim_{\eta \to 0^+} p(t = u)\mathbb{E}\Bigg[ 2\Big(-\tilde{f}^* + v\Big)^\top \Big(-g_\eta\Big) \,|\, t = u\Bigg]$$

$$= p(t = u)\mathbb{E}\Bigg[ \lim_{\eta \to 0^+} 2\Big(-\tilde{f}^* + v\Big)^\top \Big(-g_\eta\Big) \,|\, t = u\Bigg]$$

$$= p(t = u)\mathbb{E}\Bigg[ \lim_{\eta \to 0^+} 2\|-\tilde{f}^* + v\|_2^2 \,|\, t = u\Bigg]$$

This term is greater than zero by assumption that the velocity does not match at the stationary point and assumption on the positive probabiity on $p(t = u) > 0$.

Expanding the second term (with $p(t < u)$):

$$\lim_{\eta \to 0^+} p(t < u)\mathbb{E}[\phi_\eta \,|\, t < u] = \lim_{\eta \to 0^+} p(t < u)\mathbb{E}\Bigg[ 2\Big((u - t)(\partial_t \tilde{f}^*) - \tilde{f}^* + (I + (u - t)(\partial_x \tilde{f}^*))\dot{x}_t\Big)^\top$$

$$\Big((u - t)\partial_t g_\eta - g_\eta + (u - t)(\partial_x g_\eta)\dot{x}_t\Big)$$

$$- 2\Big[\Big(I + (u - t)\partial_x \tilde{f}^*\Big)(\dot{x}_t - v)\Big]^\top (u - t)\partial_x g_\eta(\dot{x}_t - v) \,|\, t < u\Bigg]$$

$$= p(t < u)\mathbb{E}\Bigg[ \lim_{\eta \to 0^+} 2\Big((u - t)(\partial_t \tilde{f}^*) - \tilde{f}^* + (I + (u - t)(\partial_x \tilde{f}^*))\dot{x}_t\Big)^\top$$

$$\Big((u - t)\partial_t g_\eta - g_\eta + (u - t)(\partial_x g_\eta)\dot{x}_t\Big)$$

$$- 2\Big[\Big(I + (u - t)\partial_x \tilde{f}^*\Big)(\dot{x}_t - v)\Big]^\top (u - t)\partial_x g_\eta(\dot{x}_t - v) \,|\, t < u\Bigg]$$

There's no $\eta$ in the first term in each of the two dot products that make up the expectand, so we can focus on the second term in the dot products, where $t < u$ For the second term in the first dot product:

$$\lim_{\eta \to 0^+} \Big((u - t)\partial_t g_\eta - g_\eta + (u - t)(\partial_x g_\eta)\dot{x}_t\Big)$$

$$= \lim_{\eta \to 0^+} \Big((u - t)\partial_t g_\eta - I_\eta(t, u)\Big(\tilde{f}^*(t, t, x) - v(t, x)\Big) + (u - t)(I_\eta(t, u)\partial_x[\Big(\tilde{f}^*(t, t, x) - v(t, x)\Big)])\dot{x}_t\Big)$$

$$= \lim_{\eta \to 0^+} (u - t)\partial_t g_\eta$$

$$= \lim_{\eta \to 0^+} (u - t)\partial_t[I_\eta(t, u)\Big(\tilde{f}^*(t, t, x) - v(t, x)\Big)]$$

$$= \lim_{\eta \to 0^+} (u - t)(\partial_t I_\eta)g + I_\eta \partial_t g$$

$$= \lim_{\eta \to 0^+} (u-t)(\partial_t I_\eta) I_\eta(t,u) \left( \tilde{f}^*(t,t,x) - v(t,x) \right)$$

$$= \left( \tilde{f}^*(t,t,x) - v(t,x) \right)(u-t) \lim_{\eta \to 0^+} (\partial_t I_\eta) I_\eta(t,u) = 0$$

The last equality holds because the function and its time derivative both go to zero.

This means the first product in the expecation goes to zero. By a similar argument the second term goes to zero as well.

Putting it all together

$$L := \lim_{\eta \to 0^+} s(\eta) = \lim_{\eta \to 0^+} \mathbb{E}[\phi_\eta] = \lim_{\eta \to 0^+} p(t=u)\mathbb{E}[\phi_\eta \,|\, t=u] + p(t<u)\mathbb{E}[\phi_\eta \,|\, t<u] > 0$$

Resultingly,

$$\exists \eta_0 > 0 \text{ s.t. } \forall \eta^* \text{ s.t. } 0 < \eta^* < \eta_0 \implies |s(\eta^*) - L| < \epsilon \tag{84}$$

If we pick $\epsilon = 0.5L$ then

$$|s(\eta^*) - L| < .5L \implies s(\eta^*) > L - .5L = .5L > 0 \tag{85}$$

so $s(\eta^*) > 0$. But this contradicts being at a stationary point. It cannot be that $\tilde{f}^*(t,t,\cdot) \neq v(t,\cdot)$ Therefore the velocity must match.

$\square$

## A.5 THEOREM 1

We present a proof about the functional stationary points of the GameFlow functional. We use the definitions of first variation and stationary point from section A.1 and the definition of functional stopgrad from section A.2.

**Theorem.** *Let $q(t, u)$ be a joint distribution with support over $t \leq u$ and with positive probability on $t = u$. Let the family $\tilde{\mathcal{F}}$ include functions $\tilde{f}$ that are continuously differentiable in all arguments. Let $x_t = \alpha_t x_0 + \sigma_t x_1$ and $\dot{x}_t = \dot{\alpha}_t x_0 + \dot{\sigma}_t x_1$. Evaluate $f$ at $f(t, u, x) + (u - t)\tilde{f}(t, u, x)$. Take expectations over $q(t, u)$ and $q(x_0)q(x_1)$. Let sg stand for stop-gradient. Define:*

$$\mathcal{L}[\tilde{f}] := \mathbb{E}\left[\|(\partial_t f)|_{(t,u,x_t)} + (\partial_x f)|_{(t,u,x_t)}\dot{x}_t\|^2 - \|(\partial_x f)_{(t,u,x_t)}(\dot{x}_t - \mathbb{E}[\dot{x}_t|x_t])\|^2\right]$$

$$\mathcal{L}^{sg}[\tilde{f}] := \mathbb{E}\left[\|(\partial_t f)_{(t,u,x_t)} + (\partial_x f)_{(t,u,x_t)}\dot{x}_t\|^2 - \|(\partial_x f)_{(t,u,x_t)}(\dot{x}_t + sg[(\partial_t f)]_{(t,t,x_t)})\|^2\right]$$

*Then $\tilde{f}^*$ is a stationary point of $\mathcal{L}^{sg}$ w.r.t. $\tilde{\mathcal{F}}$ if and only if $\tilde{f}^*$ is a stationary point of $\mathcal{L}$ w.r.t. $\tilde{\mathcal{F}}$.*

*Proof.* **Case 1: If $\tilde{f}^*$ is a stationary point of $\mathcal{L}$, then $\tilde{f}^*$ is a stationary point of $\mathcal{L}^{sg}$.**

- Since $\tilde{f}^*$ is a stationary point, $\delta\mathcal{L}[\tilde{f}^*; \cdot] = 0$

- by section A.4, we have that $\partial_t f^*(t, t, x) = -\tilde{f}^*(t, t, x) = -\mathbb{E}[\dot{x}_t|x_t]$ where $f^* = x + (u - t)\tilde{f}^*$

- $\mathcal{L}^{sg} = \mathcal{L}$

- Since $\mathcal{L}^{sg} = \mathcal{L}$, then $\delta\mathcal{L}^{sg}[\tilde{f}^*; \cdot] = \delta\mathcal{L}[\tilde{f}^*; \cdot] = 0$

**Case 2: If $f^*$ is not a stationary point of $\mathcal{L}$, then $f^*$ is not a stationary point of $\mathcal{L}^{sg}$.**

Since $f^*$ is not a stationary point of $\mathcal{L}$, then $\exists h$ that is admissible (continuous) such that $\delta\mathcal{L}[f^*; h] \neq 0$. Then,

$$\underbrace{\delta\mathcal{L}[f^*; h]}_{\text{LHS}} = \underbrace{\mathbb{E}\Big[1[t = u]\ldots\Big]}_{\text{RHS-L}} + \underbrace{\mathbb{E}\Big[1[t \neq u]\ldots\Big]}_{\text{RHS-R}} \tag{86}$$

If the LHS is nonzero, then one of RHS-L or RHS-R is nonzero. Consider both cases.

**case 2a:The RHS-R is nonzero and RHS-L is zero.** RHS-L being zero means that the velocity matches, which means that RHS-R has the same first variation between $\mathcal{L}$ and $\mathcal{L}^{sg}$. So they must coincide regarding stationary points.

**case 2b: RHS-L is nonzero and RHS-R is either zero or nonzero.**

**case2b-i** $\delta\mathcal{L}^{sg}[f^*; h] \neq 0$ directly holds. This is all we are trying to ensure anyway, so we are done in this case.

**case2b-ii** Define the soft indicator, $I_\eta$:

$$I_\eta(t, u) = 1[\eta > 0]2\Big(1 - \frac{1}{1 + \exp(-\frac{1}{\eta^2}(t - u)^2)}\Big) + 1[\eta = 0]1[t = u]. \tag{87}$$

Define the direction, $\hat{h}_\eta$:

$$\hat{h}_\eta(t, u, x) = I_\eta(t, u)h(t, u, x). \tag{88}$$

$\hat{h}_\eta$ is continuously differentiable for any $\eta > 0$. This is true cause it's a product of two functions that are each continuously differentiable ($h$ is assumed continuously differentiable). Recall the mapping $s(\eta)$ from Section A.4, that maps $\eta$ to $\delta\mathcal{L}[f; \hat{h}_\eta]$. Under the conditions of dominated convergence established in Section A.4, we know that $\exists \eta^* > 0$ such that $\hat{h}_{\eta^*}$ is a continuously differentiable function for which $\delta\mathcal{L}[f^*; \hat{h}_{\eta^*}] \neq 0$. This must mean that the velocity is not matched. But we know that if the velocity does not match, we are not at a stationary point of $\mathcal{L}^{sg}$ either, since $\mathcal{L}^{sg}$ and $\mathcal{L}$ coincide on penalizing velocity matching on $t = u$.

## B  OTHER USEFUL DERIVATIONS AND RESULTS

### B.1  GRADIENT UPDATES ARE NOT OPTIMIZATION OF ONE SCALAR OBJECTIVE VIA GRADIENTS

We show here that there exists a data distribution and a model such that the Gameflow updates are not the gradient of any single scalar objective. We illustrate this by considering an simple 1D setting. The key point is that, even in this restricted case, the update field induced by the stopgrad operator has non-zero curl and therefore cannot be written as the gradient of any scalar objective $J(\theta)$. The form to be differentiated is:

$$
\begin{aligned}
L_{\text{sg}}(\theta) = \mathbb{E}_{X_t}\left[\left\|\partial_t f_\theta(t, u, X_t) + (\partial_x f_\theta(t, u, X_t))\,\dot{X}_t\right\|^2\right] \\
- \mathbb{E}_{X_t}\left[\left\|(\partial_x f_\theta(t, u, X_t))\big(\dot{X}_t - \text{stopgrad}[\tilde{f}_\theta(t, t, X_t)]\big)\right\|^2\right].
\end{aligned}
\tag{89}
$$

for $f_\theta(t, u, x) = x + (u - t)\tilde{f}_\theta(t, u, x)$. Let us work in 1D and fix values of $X_t = x$ and $\dot{X}_t = d$, a constant. To accomplish this, we can choose $X_1$ freely and set $X_0 = X_1 - d$, so that the interpolation satisfies both $X_t = x$ and $\dot{X}_t = d$. In this case the expectations in (89) collapse to evaluation at this point (equivalently, think of us approximating with 1 Monte Carlo sample). Now, consider the parameters $\theta = (\theta_1, \theta_2)^\top$ and a single time pair $(t, u)$ such that at the position $X_t = x$,

$$
\partial_t f_\theta(t, u, x) = 0, \qquad \partial_x f_\theta(t, u, x) = \theta_1, \qquad \tilde{f}_\theta(t, t, x) = \theta_2.
\tag{90}
$$

(For a sufficiently expressive model, such local values can be realized; we only need existence of such a configuration.). Plugging these into (89) and dropping the expectation (single point), the Gameflow functional is:

$$
L_{\text{sg}} = (\theta_1 d)^2 - \big(\theta_1\big(d - \text{stopgrad}[\theta_2]\big)\big)^2.
\tag{91}
$$

Let $\tilde{\nabla}$ denote differentiation with the stopgrad applied to $\theta_2$ in the second term. Differentiating (91) with respect to $\theta_1$ yields

$$
\tilde{\nabla}_{\theta_1} L_{\text{sg}} = 2\theta_1 d^2 - 2\theta_1\big(d - \theta_2\big)^2
\tag{92}
$$

$$
= 2\theta_1\Big(d^2 - (d - \theta_2)^2\Big)
\tag{93}
$$

$$
= 2\theta_1\Big(d^2 - (d^2 - 2d\theta_2 + \theta_2^2)\Big)
\tag{94}
$$

$$
= 2\theta_1\big(2d\theta_2 - \theta_2^2\big)
\tag{95}
$$

$$
= 2\theta_1\theta_2(2d - \theta_2).
\tag{96}
$$

Here the stopgrad on $\theta_2$ only affects the second term and does not change the first term. For $\theta_2$, all occurrences appear inside a stopgrad and the first term does not depend on $\theta_2$, hence

$$
\tilde{\nabla}_{\theta_2} L_{\text{sg}} = 0.
\tag{97}
$$

Thus the update field induced by SGFlow in this simple example is

$$
g(\theta_1, \theta_2) := \big(g_1(\theta_1, \theta_2), g_2(\theta_1, \theta_2)\big) = \big(2\theta_1\theta_2(2d - \theta_2),\ 0\big).
\tag{98}
$$

If this update were the gradient of some scalar objective $J(\theta_1, \theta_2)$, then the mixed partial derivatives would commute:

$$
g_1 = \partial_{\theta_1} J, \qquad g_2 = \partial_{\theta_2} J \quad \Rightarrow \quad \partial_{\theta_2} g_1 = \partial_{\theta_1} g_2.
\tag{99}
$$

However,

$$
\frac{\partial g_1}{\partial \theta_2} = \frac{\partial}{\partial \theta_2}\Big(2\theta_1\theta_2(2d - \theta_2)\Big) = 2\theta_1\big(2d - \theta_2 - \theta_2\big) = 4\theta_1(d - \theta_2),
\tag{100}
$$

$$
\frac{\partial g_2}{\partial \theta_1} = 0,
\tag{101}
$$

and hence the curl of the update field is

$$
\frac{\partial g_1}{\partial \theta_2} - \frac{\partial g_2}{\partial \theta_1} = 4\theta_1(d - \theta_2),
\tag{102}
$$

which is non-zero for generic $\theta$ (for example, whenever $\theta_1 \neq 0$ and $\theta_2 \neq d$). Therefore $g$ is a smooth vector field with non-zero curl and *cannot* be written as the gradient of any scalar objective $J(\theta_1, \theta_2)$ in general. Put differently, a constraint on the relationship between the parameters of the model and the value of the chosen datapoint is necessary to ensure zero curl.

This 1D example is a specific instantiation of Gameflow (89) with a simple model and a single training point. It shows that, once we introduce the stopgrad on $\tilde{f}_\theta(t, t, x)$, the resulting optimization dynamics are in general *non-conservative*. The stopgrad structure breaks the symmetry required for the updates to be the gradient of a single scalar function. In this sense, SGFlow is formally a (two-player) *game* rather than standard gradient descent on one potential function. Or one prefers, a nongradient vector flow.

$\square$

### B.2 DIFFERENTIATING W.R.T. T VERSUS U

Picking one of $t$ or $u$ just switches the sign of some terms in the loss, which ultimately only affects whether the learned map goes upward in time or downwards. The PDE can be obtained by differentiating the flow map $f(t, u, x)$ with respect to either endpoint. If $X_s$ solves $\dot{X}_s = v(s, X_s)$, then keeping $u$ fixed and differentiating in $t$ gives a condition that evaluates $v$ at $t$ and $x$:

$$\partial_t f(t, u, x) + (\partial_x f(t, u, x))\, v(t, x) = 0, \tag{103}$$

while keeping $t$ fixed and differentiating in $u$ gives a condition that evaluates $v$ at $u$ and $x_u = f(t, u, x)$:

$$\partial_u f(t, u, x) + (\partial_x f(t, u, x))\, v\big(u, f(t, u, x)\big) = 0. \tag{104}$$

Along the ODE trajectory $dX = v\, ds$ one has the identity

$$\partial_t f(t, u, x) = -\, \partial_u f(t, u, x),$$

so choosing to differentiate in $t$ versus $u$ only changes the sign convention and which endpoint is held fixed.

