# OpenReview forum: "Flow Map Learning Via Non-Gradient Vector Flow"
_ICLR.cc/2026/Conference — ICLR 2026 Poster_

### Official Review · Reviewer_NL1h · 2025-10-29

**Soundness:** 3
**Presentation:** 2
**Contribution:** 3
**Rating:** 6
**Confidence:** 3

**Summary:**

The paper proposes GameFlow, an approach for learning flow maps that bypasses explicit invertibility constraints
and expensive differentiation through model iteration. It introduces a new training objective of flow-based models with theoretical guarantee of optimality in certain sense.
Numerical experiments are conducted to verify the results.

**Strengths:**

The paper is theoretically grounded and proves that the proposed loss (7) is consistent with the true loss function in the sense that they share same functional stationary points. This is a non-trivial improvement upon MeanFlow. And the theories are also verified by experiments, where the solution to MeanFlow is not the ground truth in contrast to GameFlow. In addition, GameFlow outperforms MeanFlow and other methods on real-data experiments like CIFAR-10 generation.

**Weaknesses:**

1. Although the training objective is theoretically sound, it involves much higher computational cost than MeanFlow due to the backpropagation of JVP, especially when scaling to large models. This is the most severe issue of the proposed method and is inevitable in order to make the objective mathematically correct. Could the authors conduct larger scale experiments if possible to see the scaling ability of the proposed method?

2. I don't see any connection between the proposed framework with game and the interpretation as two-player game is quite far-fetched. The authors may need to choose a more approriate title.

**Questions:**

Please see weakness part.

---

> ### Author Response · Authors · 2025-11-27
> **Thank you for your feedback and questions.**
>
> **Although the training objective is theoretically sound, it involves much higher computational cost than meanflow due to the backpropagation of JVP, especially when scaling to large models. This is the most severe issue of the proposed method and is inevitable in order to make the objective mathematically correct. Could the authors conduct larger scale experiments if possible to see the scaling ability of the proposed method?**
>
> This is queued and hopefully will finish before December 1.
>
> **I don't see any connection between the proposed framework with game and the interpretation as two-player game is quite far-fetched. The authors may need to choose a more approriate title.**
>
> We thank you for the suggestion. Please see our response to this question in the common response.

---

### Official Review · Reviewer_oX9E · 2025-10-29

**Soundness:** 3
**Presentation:** 3
**Contribution:** 3
**Rating:** 8
**Confidence:** 3

**Summary:**

The paper derives a new loss for training Flow Maps models, i.e. models that can approximate ODEs in 1 step. They derive their objective function from first principles and prove that it shares same minimizers than the ideal loss. Then, they apply their loss to toy data (multivariate Gaussian) and image generation benchmarks. In toy data setting, the demonstrate that Mean Flows objective can converge to incorrect optimum.

**Strengths:**

* Principled derivation from first principles of a novel training objective for learning Flow Maps. This is an important results, given the attention received by Flow Maps by the research community, and given their importance to reduce inference cost of diffusion models.

* Proved that the tractable training objective has same optimum than the target objective.

* Interesting analysis on the Multivariate Gaussian setting, showing that Mean Flows does not converge to the correct optimum.

**Weaknesses:**

* Experimental results seem weak and far from results in the literature. How can the authors explain this?  Indeed, 1-step models are now around 2/3 of FID on CIFAR10 (see for example "Consistency Models Made Easy" from Geng et al.). Comparing to a FID of 29 reached on 10-step sampling, this is very low.

* Weight decay is rarely used in diffusion/consistency models. Why do the authors choose to use it? Can it explain lower FID than in the rest of the literature?

**Questions:**

* Going from Equation (5) to Equation (6) is done by differentiating with regards to t. Why differentiating by t and not by u, as done in Mean Flows? Could you compare both approaches?

* You propose replacing the velocity field in (6) by sg(f). Actually you could also rely on a pre-trained neural network to approximate the velocity field. This could give another option to train your model in a distillation mode. Have you considered this?

---

> ### Author Response · Authors · 2025-11-27
> **Thank you for your feedback and questions.**
>
> **Experimental results seem weak and far from results in the literature. How can the authors explain this? Indeed, 1-step models are now around 2/3 of FID on CIFAR10 (see for example "Consistency Models Made Easy" from Geng et al.). Comparing to a FID of 29 reached on 10-step sampling, this is very low. Weight decay is rarely used in diffusion/consistency models. Why do the authors choose to use it? Can it explain lower FID than in the rest of the literature?**
>
> Thanks for pointing this out! We removed the weight decay (which we had used due to its presence in Kingma et al., Variational Diffusion Models) and the results improved for flow matching, meanflow, and gameflow. Please see the results table in the common response. However, we emphasize further differences in the meanflow numbers here from those
> in their paper and mention possible explanations in the common response.
>
> **Going from Equation (5) to Equation (6) is done by differentiating with regards to t. Why differentiating by t and not by u, as done in Mean Flows? Could you compare both approaches?**
>
> This is a good question. Picking one of $t$ or $u$ simply switches the sign of some terms in the loss, which ultimately only affects whether the learned map goes upward in time or downwards.  The PDE can be obtained by
> differentiating the flow map $f(t,u,x)$ with respect to either endpoint.
> If $X_s$ solves $\dot X_s = v(s,X_s)$, then keeping $u$ fixed and
> differentiating in $t$ gives a condition that evaluates $v$ at $t$ and $x$:
> \begin{equation}
> \partial_t f(t,u,x) + (\partial_x f(t,u,x)) v(t,x) = 0,
> \end{equation}
> while keeping $t$ fixed and differentiating in $u$ gives a condition that evaluates $v$ at
> $u$ and $x_u=f(t,u,x)$:
> \begin{equation}
> \partial_u f(t,u,x) + (\partial_x f(t,u,x)) v \big(u,f(t,u,x)\big) = 0.
> \end{equation}
> Along the ODE trajectory $dX = v\,ds$ one has the identity $\partial_t f(t,u,x) = -\partial_u f(t,u,x)$,
> so choosing to differentiate in $t$ versus $u$ only changes the sign convention and which endpoint is held fixed.
>
> What this means is that  meanflow and the first term of $L_{sg}$ from gameflow differ only in this orientation choice.
> Because changing the variable of differentiation
> introduces a minus sign, and changing from an average-velocity parameterization to its time-reversed counterpart (making the model go upward or downward) introduces a second minus sign, the two sign changes cancel.  After handle signs and converting between $\tilde{f}$ and $f = x + (u-t)\tilde{f}$, the meanflow term and the first term from gameflow's $L_{sg}$
> are mathematically equivalent. However, meanflow additionally stopgrads some of those terms while $L_{sg}$ adds a second term.
>
> We included this in appendix C.2 of the updated draft.
>
>
>
> **You propose replacing the velocity field in (6) by sg(f). Actually you could also rely on a pre-trained neural network to approximate the velocity field. This could give another option to train your model in a distillation mode. Have you considered this?**
>
> In this work, we focused on learning from scratch, but we do see the value of providing a way to use pretrained models as well.
> We could in principle take original PDE loss and evaluate it directly with a pretrained velocity $v_{\text{pt}}$.
> That is, replace
>
> $ L:= ||\partial_t f + (\partial_x f) v||^2$
>
> with
>
> $L_{\text{pt}}:= ||\partial_t f + (\partial_x f) v_{\text{pt}}||^2$.
>
> This would correspond to taking the Eulerian loss from the semi-concurrent work [1] and using a pretrained velocity.
> In contrast, we could proceed by expand the left hand side of $L$ as we did in our work, and use a pretrained velocity in place of our $\text{stopgrad}[\tilde{f}(t,t,\cdot)]$  term, which already acts as ``pretrained velocity". That is, replace
>
> $L_{\text{sg}}= ||\partial_t f + (\partial_x f) \dot x_t ||^2 - || (\partial_x f)(\dot x_t - \text{sg}[\tilde{f}(t,t,\cdot)]||^2$
>
> with
>
> $L_{\text{sg},\text{pt}} :=
>     ||\partial_t f + (\partial_x f) \dot x_t ||^2 - || (\partial_x f)(\dot x_t - v_{\text{pt}})||^2$
>
> One challenge we anticipate about pretrained velocities with either of these formulations, is that it is not enough to supervise $f(t,u,\cdot)$
> for $t<u$ with
> $v(t,x)$. We need $f(t,t,\cdot)$ to also be correct. One condition for that is for $\tilde{f}(t,t,x)$ to equal $v(t,x)$.
> But this ``copying" of the values of $v_{\text{pt}}(t,x)$ into $\tilde{f}(t,t,x)$ is itself a non-trivial learning task. Taking note of this challenge, a better strategy might be to parameterize $f=x+(u-t)\tilde{f}(t,t,x)$ with the underlying model $\tilde{f}$ directly parameterized as $\tilde{f}(t,u,x) = (1 - [u-t]) v_{\text{pt}}(t,x) + (u-t)\text{network}(t,u,x)$. We will gladly test this (and would be glad if others explore it too).
>
> [1] Boffi et al, 2025. How to build a consistency model: Learning flow maps via self-distillation. Neurips 2025.

---

### Official Review · Reviewer_mKce · 2025-11-01

**Soundness:** 2
**Presentation:** 2
**Contribution:** 2
**Rating:** 4
**Confidence:** 3

**Summary:**

This paper proposes a method, GameFlow, to learn the flow function of the PF-ODE. Unlike other flow learning alternatives, GameFlow's optimum is proven to be the true flow map, and GameFlow does not necessitate computationally expensive or restrictive operations like invertible neural networks or nested differentiation through JVPs. To this end, GameFlow leverages an objective involving only the reverse flow and the true velocity field, the latter being equivalently approximated by the stop gradient of the optimized network itself. Experiments on toy data and CIFAR-10 against other flow models illustrate the soundness and benefits of GameFlow in terms of generative performance and memory footprint.

**Strengths:**

1. GameFlow is well motivated to address the shortcomings of other flow models. Although this is not new, the paper highlights and addresses the theoretical issues of MeanFlow. Further, GameFlow alleviates the computational and design burden of otherwise sound alternatives. To my knowledge, GameFlow relies on novel derivations in the field. For these reasons, I believe that GameFlow has the potential to influence future works in this research area.
2. The proposed method seems, up to my limited assessment of the appendix, sound. Empirical results do confirm its advantages within the limits of the chosen experimental setting (cf. weaknesses).
3. The paper is overall clearly written, except for a few issues described below.

**Weaknesses:**

### Questions on main claims

1. The interpretation of GameFlow as a game is far-fetched. It solely relies on the presence of a stop gradient in the objective, making the second player trivial. Besides, while this does change the optimization landscape, this is the case of many other methods in or outside the domain (such as many the papers cited in the submission).
2. Better intuition and presentation for the main result (Theorem 1) would be appreciated. Without reviewing the appendix in detail, it is challenging the understand the rationale behind the result -- although I understand this is a challenging task. My main question regards the fact that repeated arguments (l. 215 in the main paper, l. 1375 and 1406 in the appendix) rely on the loss values to conclude on the loss gradients, which is usually inconclusive when a stop gradient operator is involved.

### Questionable experimental setting

3. GameFlow is tested on limited benchmarks. While this is understandable depending on compute constraints and not eliminatory, it would have beneficiated from a more extensive empirical study involving standard datasets like ImageNet.
4. More importantly, the FID values on CIFAR-10 are unusually high (compared to e.g. MeanFlow and other papers in the literature). This raises doubts on the realism of the experimental setting, and questions the assertion that the training configuration is "common" (l. 320).
5. Reproducibility is limited as only part of the codebase is available within the submission PDF itself, and the paper shares few details. The promise to release the source code is duly noted.

### Advantage w.r.t. consistency models

6. The paper mentions consistency models as related methods but ignores them when discussing the method's advantages. To my understanding and based on Table 1, consistency models are sound and scalable; the only difference is that they do not model the entire flow (i.e. they are not multistep as stated in the paper). The implications of this difference in practice are unclear in the paper, especially as consistency models are left out of the experimental section.

### Other issues

7. Two minor claims need to be further clarified.
    - The benefits of using split batch JVP are not explained or shown experimentally.
    - The authors state the that gradients of stop-grad' loss "are not the gradients of any one mathematical objective", but there is no proof for this. It may be that these gradients do correspond to a mathematical objective.
8. There are a number of formatting issues and typos (some of them spotted below). I would advise the authors to proofread their manuscript.
    - Table 1 would be more readable with symbols such as ✓ and ✗.
    - Parentheses are missing for the reference l. 75.
    - "covvariance" should be "covariance" (l. 176).
    - "as" should be "is" (l. 195).
    - The second $\hat{f}$ should be $\tilde{f}$ l. 215, following l. 185.
    - Figure 1 should be integrated using vector graphics.
    - There is an extra period l. 303 and an extra space l. 345.

**Questions:**

The contributions of this paper appear to be valuable for the community but are hindered by presentation issues and a questionable experimental setting. Therefore, I do not recommend acceptance in the current state of the submission. Still, depending on the authors' response to the above weaknesses, I am willing to change my assessment.

Most importantly, I suggest the authors to address the following points.
1. Remove from the method name and framing as well as the paper title the notion of game.
2. Provide further intuition on the rationale behind Theorem 1.
3. Explain the experimental difference with other papers.
4. Clarify the advantage of the method compared to consistency models.

---

> ### Author Response · Authors · 2025-11-27
> **Thank you for your feedback and questions.**
>
> **Main claims: The interpretation of gameflow as a game is far-fetched. It solely relies on the presence of a stop gradient in the objective, making the second player trivial. Besides, while this does change the optimization landscape, this is the case of many other methods in or outside the domain (such as many the papers cited in the submission).**
>
> We thank you for the suggestion. Please see our response to this question in the common response
>
> **Main claims: Better intuition and presentation for the main result (Theorem 1) would be appreciated. Without reviewing the appendix in detail, it is challenging the understand the rationale behind the result -- although I understand this is a challenging task. My main question regards the fact that repeated arguments (l. 215 in the main paper, l. 1375 and 1406 in the appendix) rely on the loss values to conclude on the loss gradients, which is usually inconclusive when a stop gradient operator is involved.**
>
> Thanks for good question and we have omitted the loosely phrased lines on ``$L_{\text{sg}} \to L$". The theorem does not rely on this loose claim. For more intuition, the main theorem shows that the optimizing of $L_{\text{sg}}$
> and $L$ stop at the same set of functions.
> There are some technically challenging steps to show this, for example using limits of soft indicators to show that the functions considered are always continuous, a necessary property of flow maps. However, the result yields something intuitive and desirable:
>
> - despite the stopgrad, $L_{\text{sg}}$ still trains the non-stopgrad version of $\tilde{f}(t,t,x)$ to match $v(t,x)$; in other words, the velocity term inside the stopgrad is trained by the rest of the loss.
> - $L_{\text{sg}}$ is not at a stationary point when this velocity estimate is inaccurate, so the optimization continues moving and does not become stuck at functions that distill an incorrect velocity.
>
> Slightly more technically,  the theorem shows two things
>
> - $L_{sg}$ favors all the solutions that $L$ does
> - $L_{sg}$ does not favor any other solutions
>
> More precisely, about your specific concerns with $L_{sg}\to L$
> - we show that the loss $L$ with true velocity $v$ only has stationary points at functions equal to the sought-after flow map $f$.
> - Let the model be parameterized as  $f_\theta = x + (u-t)\tilde{f}$,
> - We show that if $\tilde{f}(t,t,\cdot) \neq v(t,\cdot)$ then $f_\theta$ is not a stationary point of $L$
> - We show that any function that is not a stationary point of $L$ is also not a stationary point of $L_{sg}$
> - Together, the last two points mean that  $L_{sg}$ does not stop at places where $\tilde{f}(t,t,\cdot) \neq v(t,\cdot)$
> - Instead, we show that $L_{sg}$ only stops at places where $\tilde{f}(t,t,\cdot) = v(t,\cdot)$
> - But in those places, $L_{sg} = L$
> - So any other conditions that $L$ imposes on its stationary points are also imposed by $L_{\text{sg}}$ (and only those conditions are imposed).
>
> The main reason all of this works out is that $\tilde{f}(t,t,\cdot)$ gets trained toward the quantity it should equal (here the velocity $v$) via its appearance in other terms outside of the stopgrad. This other terms play a crucial role, and not all stopgrad optimizations  benefit from this property. All of this is shown with proof and not reliance on the empirical condition that $L_{\text{sg}}$ approaches $L$.
>
> **Experimental: gameflow is tested on limited benchmarks. While this is understandable depending on compute constraints and not eliminatory, it would have benfitted from a more extensive empirical study involving standard datasets like ImageNet.**
>
> This is queued and hopefully will finish before dec 1.
>
> **Experimental: More importantly, the FID values on CIFAR-10 are unusually high (compared to e.g. meanflow and other papers in the literature). This raises doubts on the realism of the experimental setting, and questions the assertion that the training configuration is "common" (l. 320).**
>
> Please refer to our response on cifar-10 meanflow benchmarking in the common response. As for common configuration: By common we meant that we use the OpenAI Unet [1] using channel multipliers (1,2,2,2) similarly to Flow Matching,
> and LR=2e-4, similar to the DiT line of works.  We should specify that common refers to “common for flow matching + diffusion models”, but not common for flow maps because only a few works have so far studied this learning task (learning the integral of the function normally learned in flow matching) and each work currently does something different.
>
> [1]  [Github Repository](https://github.com/openai/guided-diffusion}{(Github repository))

---

> > ### Author Response · Authors · 2025-11-27
> > **(Response Continued)**
> >
> > **Experimental: Reproducibility is limited as only part of the codebase is available within the submission PDF itself, and the paper shares few details. The promise to release the source code is duly noted.**
> >
> > Please see the common reply above for a link to a simple version of the code. We will include a fuller version by the end of the discussion period.
> >
> > **Consistency: The paper mentions consistency models as related methods but ignores them when discussing the method's advantages. To my understanding and based on Table 1, consistency models are sound and scalable; the only difference is that they do not model the entire flow (i.e. they are not multistep as stated in the paper). The implications of this difference in practice are unclear in the paper, especially as consistency models are left out of the experimental section.**
> >
> > CTM points out shortcomings of consistency models. Highlighting the issue with methods featuring 1 time argument,
> > the CTM authors (Kim et al, 2023) note that this CM multistep sampling approach (Song et al., 2023) ``exhibits degrading sample quality with increasing NFE, lacking a clear trade-off between computational budget (NFE) and sample fidelity".
> >
> > Please see our extended answer on this topic to reviewer peHa (above). We summarize that answer here:
> >
> > Our interest is in flow map learning. The CTM loss has 3 terms that they use to learn flow maps.
> > CTMs are well-engineered with [complex code](https://github.com/sony/ctm/tree/51fdb914f6761398178970929804da0e6bae5c23/code). The last term, the GAN term, can improve any objective and is sufficient to match a data distribution itself.
> > The first and second terms are the flow map terms. We see in the gaussian example that CTMs' first and second term do not yield a good fit. A notable detail is that CTMs have several chains of stopgrads, and second-order integration during training. Nothing is proved about the stopgrad update rules or the discretization error's effect on the objective. These details are probably why the method does not work on Gaussians, since in theory, without stopgrads or numerical integration, the math is sound.
> >
> > **The benefits of using split batch JVP are not explained or shown experimentally.**
> >
> > Thanks for your question. We have clarified this in the text. We don't advocate specifically for any benefit of this splitting. Rather, we have a loss that features two JVPs and we unbiasedly estimate this loss by splitting the batch into half, one for each JVP.
> >
> > **The authors state the that gradients of stop-grad' loss "are not the gradients of any one mathematical objective", but there is no proof for this. It may be that these gradients do correspond to a mathematical objective.**
> >
> > This is a good question. We thank the reviewer for pointing out more places for us to add details. We have added a proof of this in section C.1 of the appendix of the updated draft. The main technique is to show that the gradient updates resulting from $\mathcal{L}_{\text{sg}}$ have non-zero curl, meaning the system is non-conservative and thus not the gradient of any scalar objective.
> >
> >
> > **Formatting/typo issues**
> >
> > Thank you very much for the close reading and formatting/typo catches, which we have fixed in our draft.

---

> > > ### Author Response · Authors · 2025-11-27
> > > **(Response Continued)**
> > >
> > > **Remove from the method name and framing as well as the paper title the notion of game.**
> > >
> > > We value the reviewer's feedback and would be happy to change the title to something like ``Using stopgrads to learn flow maps with provable stationary points" and edit out the game language. We have made this preliminary change in the draft.
> > >
> > > **Provide further intuition on the rationale behind Theorem 1.**
> > >
> > > The main point of the theorem is to establish that $L_{\text{sg}}$ (which uses a stopgrad on $\tilde{f}(t,t,x)$) has the same set of solutions as $L$ (which uses the true velocity $v$). The intuition is that when $t=u$, $L_{\text{sg}}$ tries to match the velocity. The challenge is that this signal (non zero vector field value/perturbation direction) may either be washed out by the $t<u$ terms or it may not even be a perturbation direction (i.e., not a continuous function). The theorem establishes that this type of cancellation is not possible even when restricted to continuous functions.
> > >
> > > **Explain the experimental difference with other papers.**
> > >
> > > While the results have improved due to reviewers' suggestions, the differences that remain are the long training times and high learning rates facilitated by large batch sizes, which are facilitated by access to more compute and engineering resources.
> > >
> > > **Clarify the advantage of the method compared to consistency models.**
> > >
> > > The CTM paper and Flow Map Matching papers points out that CMs have been shown to not have a monotonic tradeoff between number of inference steps and sample quality, and makes the case for two-time maps that allow multistep sampling without heuristic re-noising. As for CTM, their method relies on an extra GAN loss, orthogonal to the flow map loss, to stabilize training (we showed in the responses here that the method can struggle even to fit a 2D gaussian without the GAN loss). Such GAN losses can in principle be used to improve many kinds of generative models, so we stick to studying the core flow map loss of CTM. The method we present avoids some of the choices that may lead to such instability, like optimizing through the nesting of model calls, and we show the ability to train on Gaussians and cifar-10 without additional GAN losses.

---

### Official Review · Reviewer_PeHa · 2025-11-01

**Soundness:** 2
**Presentation:** 3
**Contribution:** 3
**Rating:** 4
**Confidence:** 3

**Summary:**

This paper proposes GameFlow, an objective for training flow map models for fast sampling. The method simultaneously trains the model to predict both the flow map and its underlying velocity, using a self-correcting stop-gradient target. This approach avoids costly model inversions and large Jacobians. The method is supported by a theoretical proof of correctness and empirical results on cifar-10.

**Strengths:**

1. Theorem 1 rigorously proves that the true flow map is a stationary point of the GameFlow objective. The comprehensive proofs are technically solid.
2. Provides reasoning and numerical evidence (Figure 1) that MeanFlow does not preserve the true flow map as an optimum.
3. The paper is well-written and clearly motivated. It systematically identifies and addresses key challenges in the field.

**Weaknesses:**

1. Only cifar-10 is evaluated with no higher-resolution datasets. According to line 308, the cifar-10 experiments are conditional, yet MeanFlow at 100 steps achieves FID 4.58, which is worse than the 2.92 reported in Table 3 of the MeanFlow paper[1] (Unconditional cifar-10, NFE=1). This discrepancy needs clarification.
2. Missing comparisons with consistency model variants (Consistency Models[2], simplified Consistency Models[3], etc.).
3. Missing training curves (loss, FID vs. steps) to verify training stability and convergence behavior.
4. What is the empirical impact of the corrective term in the loss? An ablation study is needed.

[1] Geng, Zhengyang, et al. "Mean flows for one-step generative modeling." arXiv preprint arXiv:2505.13447 (2025).

[2] Song, Yang, et al. "Consistency models." (2023).

[3] Lu, Cheng, and Yang Song. "Simplifying, stabilizing and scaling continuous-time consistency models." arXiv preprint arXiv:2410.11081 (2024).

**Questions:**

1. Can you provide a theoretical comparison between GameFlow and Consistency Trajectory Models[1]?
2. What specifically causes the 3× memory overhead versus MeanFlow when both methods use JVPs?

[1] Kim, Dongjun, et al. "Consistency trajectory models: Learning probability flow ode trajectory of diffusion." arXiv preprint arXiv:2310.02279 (2023).

---

> ### Author Response · Authors · 2025-11-27
> **Thank you for your feedback and questions.**
>
> **Only cifar-10 is evaluated with no higher-resolution datasets. According to line 308, the cifar-10 experiments are conditional, yet meanflow at 100 steps achieves FID 4.58, which is worse than the 2.92 reported in Table 3 of the meanflow paper[1] (Unconditional cifar-10, NFE=1). This discrepancy needs clarification.**
>
> Please refer to our response on cifar-10 meanflow benchmarking in the common response.
>
> **Missing comparisons with consistency model variants**
>
> CTMs make arguments for two time methods over one-time methods like consistency models. We mention some points here, and describe the theoretical connection between CTMs and gameflow. We then run the method on a low-dimensional example.
>
> - Consistency Models (CMs) use only a single time argument and therefore cannot express arbitrary two-time flow maps; to take more than one sampling step, they rely on a re-noising construction that maps noisy $x_t$ to data estimate $x_0$  and then pushes data estimate $x_0$ back up to noisy $x_s$ for $0 < s < t$ using the forward noising process. While this allows multistep sampling, the re-noising step necessarily takes the trajectory off the probability-flow ODE and the resulting updates no longer correspond to integrating the PF-ODE solution. By contrast, the methods we benchmark against flow-map matching (LSD/PSD/ESD) and meanflow, are two-time methods that hope to model the true flow map that corresponds to the PF-ODE trajectory, throughout inference. Because our work is specifically about learning two-time PF-ODE flow maps and comparing methods that share this goal and structure, we benchmark against other two-time map-matching approaches rather than one-time consistency models whose sampling dynamics are qualitatively different and not designed to approximate the PF-ODE flow.
>
> - *Importance of two times.* Highlighting the issue with methods featuring 1 time argument, the CTM authors (Kim et al, 2023) note that  this CM multistep sampling approach (Song et al., 2023) ``exhibits degrading sample quality with increasing NFE, lacking a clear trade-off between computational budget (NFE) and sample fidelity"
>
> - *A brief review of the CTM method.*  CTM trains a model $G(x_u, u, t)$ to bring $x_u$ down to time $t$ where $x_0$ is data. CTMs use 3 losses: the consistency loss, a denoising loss, and a GAN loss. The first is derived from the mathematical properties of flow maps and the 2nd is related to making sure the model matches the velocity itself. The GAN loss helps the output distributions at each time $t$ match  their correct marginal distribution.  The main loss itself also involves a fairly complicated nesting of models as well as velocity-integration-during training.
>   - let $0 < t < m < u < 1$ where for CTM, $x_0$ is data.
>   - the teacher model maps noisy $x_u$ down to $x_m$ by integrating the current (stopgrad) velocity estimate from $u$ down to $m$. The teacher then using the current (stopgrad) flow map estimate to jump the $x_m$ estimate down to an $x_t$ estimate. The teacher then maps the $x_t$ estimate down to an $x_0$ estimate, again by jumping with the (stopgrad) model. Altogether, we can write the teacher term as  $\text{sg}[G](t \to 0) \circ \text{sg}[G](m \to t) \circ \text{sg}[G](u \to m)(x_u)$ where the first $u \to m$ is implemented with second order integration of the velocity.
>   - The student jumps from $x_u$ down to an $x_t$ estimate with the live model, and then jumps $x_t$ down to an $x_0$ estimate,
>         through the stopgrad/frozen model.
>         \item We can write the student as
>         $\text{sg}[G](t \to 0) \circ G(u \to t)(x_u)$
>   - the main CTM loss enforces that these two $x_0$ estimates agree.
>   - the velocity matching loss makes sure that a certain derivative of $G$ matches the denoiser $\mathbb{E}[x_0|x_u]$, which makes the velocity correct.
>   - Finally, a GAN loss makes sure that the teacher and student $x_0|x_u$ are similar.
> Even without the GAN loss, there are several nested model evaluations, making gradient propagation tricky.
>
> - *Why we did not benchmark against CTM versus some of the other methods like meanflow, on the image experiments*: we believe that CTM can be made to work well, but with an immense engineering effort (gradient-norm based GAN loss weighting, and second order integrators for the teacher velocity can be seen [here](https://github.com/sony/ctm/blob/main/code/cm/train_util.py) and [here](https://github.com/sony/ctm/blob/51fdb914f6761398178970929804da0e6bae5c23/code/cm/karras_diffusion.py#L439)
> that is complementary but not intrinsically part of the flow map losses under study.

---

> ### Author Response · Authors · 2025-11-27
> **(response continued)**
>
> (continuing on CTM response)
>
> - *Empirical evaluation:* On a simple 2D gaussian example (also used below to answer the question on the corrective loss term $L_2$ in the gameflow $L=L_1+L_2$), we show samples from the first 10,000 training steps
> of CTM. We exclude the extra GAN term under the reasoning that the consistency and denoising losses together
> should be sufficient to learn the flow map in this low-dimensional example (we already know that GANs can learn Gaussians).
> Additionally, GANs can improve any generative modeling approach and is not specific to the CTM method (used
> by the same authors to improve diffusion models (Kim et al. Refining generative process with discriminator guidance in score-based diffusion models. ICML 2023). [See here for CTM plot.](https://drive.google.com/file/d/10nBjGb7I827kfY41sopnwbOfUsM1lb0U/view?usp=sharing). We see that the samples get to approximately the correct location, but a slow process is underway to stretch the samples out to the right distribution, indicating either a wrong optima or slow gradient propagation. We see that it gets closer, but still oscillates, at steps 45,000-50,000. Since the CTM theory is sound, we believe this has to do with optimizing through nested model evaluations,  which the LSD/ESD/PSD losses also do, but which gameflow avoids purposefully.
>
> - *In summary*: Our interest is in flow map learning.
> The CTM loss has 3 terms that they use to learn flow maps. CTMs are well-engineered with [complex code](https://github.com/sony/ctm/tree/51fdb914f6761398178970929804da0e6bae5c23/code). The last term, the GAN term, can improve any objective and is sufficient to match a data distribution itself. The first and second terms are the flow map terms. We see in the Gaussian example that CTMs' first and second term do not yield a good fit. A notable detail is that CTMs have several chains of stopgrads, and second-order integration during training. Nothing is proved about the stopgrad update rules or the discretization error's effect on the objective. These details are probably why the method does not work on Gaussians, since in theory, without stopgrads or numerical integration, the math is sound.
>
>
>
> **What is the empirical impact of the corrective term in the loss? An ablation study is needed.**
>
> Thank you for your suggestion. Without $L_2$, the stationary point does not correspond to the true distribution. We ran this on a 2D gaussian and observe that it does not work. Intuitively, the corrective term is needed to adjust for the stochasticity in noisy velocities, so the noise doesn't drive the flow map elsewhere
>
> $$L=\mathbb{E}[\|(\partial_t f) + (\partial_x f)v \dot X_t \|^2] - \mathbb{E}[\|(\partial_x f)(\dot X_t- v) \|^2]$$
>
> where the first term is $L_1$ and the second is $L_2$. In the main text, we had included figure 1 to illustrate this point, where the loss contours are those of $L_1+L_2$ while the minimum of $L_1$ is shown to be misplaced relative to the true flow map. To create the figure, we evaluated $L_1$ and $L_1+L_2$ on a grid of values including the true data-generating parameters.
> To show this empirically with optimization, rather than plotting all loss values, we run training with $L=L_1+L_2$ versus just $L_1$, on a simple 2D gaussian example for 7000 training steps (plenty for a 2D gaussian). With just $L_1$ on the top and the full $L$ on the bottom, we see that the training oscillates and does not settle on the desired optimum.
>
> [See here for plot](https://drive.google.com/file/d/126lvYjTc6J6iew-6qMbX1TVRDIxduw6m/view?usp=sharing)
>
> On a harder mixture-of-steep-curvy-bananas problem we also compare meanflow and gameflow. meanflow corresponds to taking just $L_1$ and stopgrad'ing all derivatives in the loss. gameflow corresponds to taking $L=L_1+L_2$ but substituting the velocity in $L_2$ with a stopgrad of the model's current velocity estimate (which is also getting training with non-stopgrad via the $L_1$ term when $t=u$).
>
> [See here for plot](https://drive.google.com/file/d/1_X945OMl7vsgrGAyNXj3oML74zHuDDJl/view?usp=sharing)

---

> > ### Author Response · Authors · 2025-11-27
> > **(Response continued)**
> >
> > **Can you provide a theoretical comparison between gameflow and Consistency Trajectory Models[1]?**
> >
> > CTM and gameflow both target the same mathematical object, the probability--flow ODE flow map (i.e., the integral of the ODE), but they learn this map through different means. CTM learns the map by distilling a teacher solver, and the losses for teacher and student involve several nested model evaluations (the teacher integrates from $t$ to $u$, then jumps from $u$ to $s$, then from $s$ to $0$; and the student jumps from $t$ to $s$ and then to $0$). The objective also depends on a chosen feature-space distance and, in practice, includes DSM and GAN terms that further influence the optimum. Consequently, the CTM loss is sensitive to the quality of the ODE discretization used by the teacher (in practice CTM finds the need to use a 2nd order solver during training) and necessitates the presence of the GAN. GANs can, in principle, be used to augment any generative model or even solve the problem itself.
> >
> > In an idealized limit with perfect teacher solves of a perfect velocity field, unlimited model capacity, perfect min--max optimization, and no auxiliary losses, the true flow map is a solution of the CTM objective because the student matches the teacher everywhere, which matches the ODE. CTM, however, does not show that this solution is unique or that its stationary points during optimization coincide with those of the underlying PDE.  Gameflow, in contrast, derives its loss directly from the flow map PDE  and proves that the stop-gradient optimization has stationary points if and only if the model equals the true flow map.
> >
> >
> > **What specifically causes the 3× memory overhead versus meanflow when both methods use JVPs?**
> >
> > The reviewer is correct that both methods use a JVP that should take equal memory in the forward pass. The difference comes from two distinctions, both arising from the stopgrad on the JVP in meanflow. First, because meanflow anticipates not needing to differentiate this term,  the JVP is run with create graph = False which avoids storing the computational graph of JVP and only keeps the end result, treated as a constant. Second, during backward, autograd does not need to differentiate such a graph. While saving memory and compute, this choice means that there is no guarantee on the stationary points of this optimization, without further proof.
> >
> >
> > [1] Kim, Dongjun, et al. "Consistency trajectory models: Learning probability flow ode trajectory of diffusion." arXiv preprint arXiv:2310.02279 (2023).

---

### Author Response · Authors · 2025-11-27
**Common Response: Thank you to the reviewers.**

We thank the reviewers for their thoughtful and detailed feedback.

We first mention a few highlights. We then answer a few common questions, provide updated results, and expand on differences with experiments from the existing works. We then proceed with reviewer-specific responses below as replies to the reviews.

**Highlights:** Reviewers highlighted that Theorem 1 “rigorously proves” that the true flow map is a stationary point of the proposed objective and described the accompanying proofs as “technically solid” (PeHa). They also noted that the paper provides a “first principles” derivation of a training objective whose tractable form shares the same optimum as the target loss (oX9E, NL1h). Several reviewers emphasized  that the work provides both theoretical reasoning and numerical evidence of challenges faced by existing methods, including the multivariate Gaussian example where meanflow does not in general converge to the correct optimum (PeHa, oX9E, mKce). Reviewers  remarked that the paper is “well-written”  and that the proposed approach is “well motivated” with derivations viewed as novel within this line of work (PeHa, mKce). Some reviewers also noted that the formulation alleviates computational and design burdens present in existing alternatives and may have potential to influence future work (mKce). Finally, reviewers observed that the empirical results are consistent with the theory, showing that gameflow recovers the correct solution where meanflow does not and performs well on CIFAR-10 within the moderate-scale experimental setting considered in the paper (NL1h, mKce).

---

> ### Author Response · Authors · 2025-11-27
> **Common Response (Continued)**
>
> # Note on name change
>
> As requested by the reviewers we have removed mention of  games for now and refer to the paper as  **Flow Map Learning via Nongradient Vector Flow**. We call the method **CurlFlow** for now in the draft,
> since the optimization follows non-conservative dynamics (i.e., those with *curl*). However, in the rest of this review response, we refer to the method by its original name, GameFlow, to not cause confusion within reviews. Please refer to it by either.
>
>
>
>
> # Updated Cifar Results:
>
> | **Method**     | **10 steps** | **50 steps** | **100 steps** | **Theory** |
> |----------------|--------------|--------------|---------------|------------|
> | Flow Matching  | 24.87        | 3.53         | 3.05          | yes        |
> | Lagrange       | 248.76       | 230.43       | 221.22        | yes        |
> | Euler          | 77.19        | 66.99        | 38.95         | yes        |
> | Progressive    | 337.36       | 235.20       | 206.18        | yes        |
> | Meanflow       | 37.32        | 4.54         | 4.23          | no         |
> | **Gameflow**   | **12.26**    | **2.88**     | **2.81**      | **yes**    |
>
> **Table:** Updated results on CIFAR after removing weight decay, training unconditionally, and training up to 200k gradient steps.
> The **theory** column indicates whether the stationary points of the optimization have been proven to exist *if and only if* the function is the flow map that integrates the ODE.
>
>
>
>
> # Differences from cifar10 results in meanflow paper
>
> We were able to find that (1) no weight decay (2) training unconditionally leads to better CIFAR FIDs. Gameflow, meanflow, and the flow matching baseline all improved from these changes. We make the following observations
>
> - Meanflow trained longer (800k steps) than we did (200k steps) with our available compute, at larger batch sizes (8x128 vs 4x128). Due to larger batch size, they also train at larger LR 6e-4 instead of 2e-4.
>
> - the meanflow paper acknowledges several non-authors from the IBM-Watson lab for helping with engineering/experimenting. Thus they benefited from an additional non-author engineering team and external compute to produce the results.
>
> - As a side note, other people had trouble replicating meanflow and mentioned that checkpoints from adjacent gradient steps can have very different samples (like all blank images, [twitter thread](https://x.com/SwayStar123/status/1933929501538619858?s=20)). This trouble highlights the important of an procedure that provably works.
>
> To summarize these points, we feel that our results for the meanflow method (academic study with out-of-box OpenAI Unet [1] batch sizes for 4 gpus, no additional non-author engineers) are as much of a datapoint about the meanflow *method* as the results in the meanflow *paper*, which are impressive and certainly tuned with state-of-art in mind  and which make use of plentiful resources. To build confidence, we include a public version of our code (see later in the common answer)
>
> [1] [OpenAI Github Repo](https://github.com/openai/guided-diffusion)
>
> |               | **Theory** | **2D Gaussian** | **Image** | **Has been tuned** |
> |---------------|------------|------------------|-----------|---------------------|
> | **Meanflow**  | no         | no               | yes       | yes                 |
> | **Gameflow**  | yes        | yes              | yes       | no                  |
>
>
>
>
> # Game terminology is trivial/far-fetched
>
> We wanted to highlight that putting stopgrad changes the update rule from gradients of a loss to something else (see response below to reviewer mKce on formal statement that the update rule is not a gradient of a scalar loss).
>  Games are the most common setting where there are update rules rather than losses.  We agree the connection is trivial, but it highlights the change from gradient following to generic update rules. It's this change away from gradients of losses (that have nice properties) that have made several of the existing methods brittle (see above comment about meanflow).
>  This change is also what necessitated our proofs on the stationary points of this update rule, derived from $L_{\text{sg}}$, match the overall stationary points derived from $L$. We value the reviewers feedback and would be happy to change the title.
>
>  For now, we pick the title **Flow Map Learning via Non-gradient Vector Flow**
>  and we call the method **CurlFlow** (since it follows update vectors that are non-conservative (i.e., have curl) We edited out the game language in the updated draft.
>
>
> # Code
>
> We will present an anonymous version of the whole code base for CIFAR by the end of the discussion period, we include a simpler repository here, which generates the 2D gaussian example used in a few places in the review answers. This code base contains gameflow , meanflow, and CTM and runs on 1 GPU.
>
> [Here is the anonymous google drive link.](https://drive.google.com/drive/folders/1j8U-Fu9H3RRBtI-9ZHF9QJrZo-dV5X3N?usp=sharing).

---

### Meta-Review · Area_Chair_6ux3 · 2026-01-06

**Summary:**

This paper proposes CurlFlow (renamed from GameFlow), a method for learning probability-flow ODE flow maps with provable correctness. The key theoretical contribution is Theorem 1, which establishes that the proposed nongradient objective has stationary points if and only if the learned map equals the true flow map, despite the use of stop-gradient terms. Reviewers generally agreed that this theoretical analysis is rigorous and clarifies important limitations of prior methods such as MeanFlow, including concrete failure modes demonstrated on correlated Gaussian examples.

**Reviewer Concerns:**

Empirical results on CIFAR-10 and toy problems support the method’s soundness, though the experimental scope is limited. Several reviewers raised concerns about the realism of the CIFAR-10 setting and discrepancies with previously reported MeanFlow results; these were addressed in the rebuttal by clarifying training choices and providing updated, more consistent baselines under controlled, academic-scale compute. Reviewers also questioned the lack of comparisons to consistency models and CTMs; the authors reasonably argue these methods target different objectives or rely on additional machinery (e.g., re-noising, teacher distillation, GAN losses), making them less directly comparable.

Overall, the main contribution is theoretical rather than empirical. While stronger empirical validation would improve the paper, the concerns raised do not undermine correctness. I view this as a sound, theory-driven contribution with moderate empirical support.

**Reviewer Scores:**

Both Reviewer PeHa and Reviewer mKce are likely to raise their score given that most of their concerns are adequately addressed.
Reviewer oX9E and Reviewer NL1h are likely to keep their positive scores.

---

### Decision · Program_Chairs · 2026-01-26

Accept (Poster)